

# Dimethyl sulfide and its role in aerosol formation and growth in the Arctic summer – a modelling study

Roya Ghahreman[1], Wanmin Gong[1], Martí Galí[2], Ann-Lise Norman[3], Stephen R. Beagley[1],

Ayodeji Akingunola[1], Qiong Zheng[1], Alexandru Lupu[1], Martine Lizotte[2], Maurice Levasseur[2], W. Richard Leaitch[1].

[1]Environment and Climate Change Canada, Toronto, Ontario, Canada

[2]Québec-Océan, Department of Biology, Université Laval, Québec, Canada

[3]Department of Physics and Astronomy, University of Calgary, Calgary, Canada

*Correspondence to*: Roya Ghahreman (roya.ghahreman@canada.ca)

**Abstract.** Atmospheric dimethyl sulfide, DMS(g), is a climatically important sulfur compound and is the main source of biogenic sulfate aerosol in the Arctic atmosphere. DMS(g) production and emission to the atmosphere increase during the summer due to greater ice-free sea surface and higher biological activity. We implemented DMS(g) in the GEM-MACH model (GEM: Global Environmental Multiscale – Environment and Climate Change Canada's (ECCC) numerical weather forecast model, MACH: ECCC's Modelling Air quality and CHemistry – chemistry and aerosol microphysics) for the Arctic region, and compared model simulations with DMS(g) measurements made in Baffin Bay and the Canadian Arctic Archipelago in July and August 2014. Two sea water DMS(aq) datasets were used as input to the simulations: 1) DMS(aq) climatology dataset based on seawater concentration measurements (Lana et al., 2011) and 2) DMS(aq) dataset based on satellite detection (Galí et al., 2018). In general, GEM-MACH simulations underpredict DMS(g) measurements, likely due to negative biases in both DMS(aq) datasets. Yet, higher correlation and smaller bias were obtained with the satellite dataset. Agreement with the observations improved by replacing climatological values with in situ measured DMS(aq) concurrently with atmospheric observations over Baffin Bay and Lancaster Sound area in July 2014.

The addition of DMS(g) to the GEM-MACH model resulted in a significant increase in atmospheric $SO_2$ for some regions in the Canadian Arctic (up to 100%). Analysis of the size-segregated sulfate aerosol in the model shows that a significant increase in sulfate mass occurs for particles with a diameter smaller than 200 nm due to formation and growth of biogenic aerosol at high latitudes (> 70º N). The enhancement in sulfate particles is most significant in the size range of 50 to 100 nm, however, this enhancement is stronger in the 200–1000 nm size range at lower latitudes (< 70º N). These results emphasise the important role of DMS(g) in the formation and growth of fine and ultrafine sulfate-containing particles in the Arctic during the month of July.





## 1. Introduction

The atmospheric aerosol plays a crucial role in climate change. Aerosol particles influence climate by absorption/scattering of short/long wave radiation (direct effect) and by changing the number/size of cloud droplets and altering precipitation efficiency (indirect effect) (e.g. Haywood and Boucher, 2000). Despite their importance in the atmosphere, there are many uncertainties and a lack of information/understanding in the estimation of their sources, composition, distribution and effects. These uncertainties are greater in the Arctic than at lower latitudes, due to the harsh environment of the Arctic that limits measurements and observations in this remote region (Bates et al., 1987).

The Arctic Ocean is an important source of gases and primary aerosols emitted into the atmosphere by gas exchange at the sea-air interface, bubble bursting and sea spray (e.g. Bates et al., 1987; Andreae, 1990; Yin et al., 1990; Leck and Bigg, 2005a, b; Barnes et al., 2006; Ayers and Cainey, 2007; Sharma et al., 2012). These emissions contain primary particles (such as sea spray) and gases, which may form secondary particles such as sulfate. Sulfate aerosols in the Arctic atmosphere originate from three main sources: anthropogenic, sea salt, and biogenic sources (Norman et al., 1999; Chang et al., 2011; Rempillo et al., 2011). Anthropogenic sulfate particles are transported into the Arctic from southern latitudes during winter and spring (Sirois and Barrie, 1999; Stone et al., 2014). During summer, wet scavenging significantly reduces anthropogenic contributions (e.g. Garrett et al., 2011; Croft et al., 2016a). Sea salt enters to the atmosphere via sea spray and bubble bursting, and is generally found in coarse mode particles (Quinn et al., 2015). The focus of this study is the main biogenic source of sulfate aerosols in the Arctic, dimethyl sulfide (DMS, with the chemical formula $(CH_3)_2S$).

During summer, DMS(aq) production and emission to the atmosphere increase due to a larger ice-free sea surface and higher bioactivities (Sharma et al., 2012; Levasseur, 2013). According to the CLAW hypothesis (Charlson et al., 1987), a negative feedback between DMS(g) emission from phytoplankton and cloud albedo can potentially regulate temperature and affect climate change over the oceans. On the global scale, the CLAW hypothesis may be flawed, particularly concerning the sea-air exchange of DMS(g) and its role in new particle formation that may subsequently impact cloud microphysics (Quinn and Bates, 2011). However, recent atmospheric observation and modeling studies suggest a significant role for DMS(g) in particle formation above oceans, especially in remote areas with low concentrations of pre-existing aerosol such as the Arctic Ocean in summer (Leaitch et al., 2013; Ghahremaninezhad et al., 2016; Quinn et al., 2017).

DMS(aq) is produced by the breakdown of dimethylsulfoniopropionate (DMSP), a compound synthesized by phytoplankton (Keller et al., 1989; Alcolombri et al., 2015). DMSP breakdown is favoured by microbial interactions and environmental stressors, and is carried out by phytoplankton and bacterial DMSP-lyase enzymes. DMS produced in the upper mixed layer of the ocean is mostly removed by bacteria and photochemistry, and only ten to fifteen percent enters into the atmosphere in the form of DMS(g) (Galí and Simó, 2015) via processes such as turbulence, diffusion and advection (Lunden et al., 2010).



The most important reaction of DMS(g) in the atmosphere is oxidation by hydroxyl and/or nitrate radicals. The addition pathway proceeds with adding the OH radical to DMS(g); its final products are dimethylsulfoxide (DMSO), dimethylsulfone ($DMSO_2$) and methanesulfinic acid (MSIA), which are all highly soluble in water and readily condensable on existing aerosols. Oxidation of DMSO, $DMSO_2$ and MSIA in the gas phase is not significant in the presence of clouds or high concentration of

aerosols (von Glasow and Crutzen, 2004). The abstraction pathway starts with the DMS(g) reaction with OH and $NO_3$ radicals, and the main products are methanesulfonic acid (MSA) and $SO_2$. The addition and abstraction pathways of DMS(g) oxidation with OH are temperature and light dependent. For example, the abstraction pathway (with the ratio of 75% of total OH and DMS oxidation) is the dominant reaction at 300 K (Hynes et al., 1986).

     The reaction of MSIA with OH may also lead to formation of $SO_2$ which could be considered as the "crossover point" between

the addition and abstraction pathways (von Glasow and Crutzen, 2004). In addition, reaction with halogens is a potential additional sink for DMS(g) in the remote marine atmosphere (von Glasow et al., 2004; Hoffmann et al., 2016). von Glasow and Crutzen (2004) focused on the oxidation of DMS(g) by halogens in the marine boundary layer (MBL) with a one-dimensional numerical model and reported significant uncertainty about the production of DMS oxidation. Hoffmann et al. (2016) included the multiphase DMS(g) chemistry in a box model. They highlighted the role of aqueous-phase DMS(g)

chemistry in the reduction of $SO_2$ and increasing MSA production. Under certain conditions and in the presence of both amines and water vapor, MSA may form new particles (Dawson et al., 2012).

     $SO_2$ derived from DMS(g) can oxidise to sulfuric acid ($H_2SO_4$), which plays an important role in particle formation and growth. Particles composed of sulfate are relatively efficient cloud condensation nuclei (CCN), which can influence the properties of clouds. Sanchez et al. (2018) highlighted the importance of phytoplankton-produced DMS emissions in CCN budget in the

Northern Atlantic. They estimated the contribution of new sulfate particles in CCN budget to be equal to 31% and 33% at 0.1% supersaturation in late-autumn and late-spring, respectively. The formation of new particles that can lead to an increase in CCN is particularly important in the Arctic during summer, when anthropogenic aerosols are scarce and the condensation sink is low (Abbatt et al., 2019; Croft et al., 2019; Leaitch et al. 2013; Ghahremaninezhad et al., 2016; Burkart et al., 2017; Collins et al., 2017).

Under the research consortium NETCARE (Network on Climate and Aerosols: Addressing Key Uncertainties in Remote Canadian Environments, Abbatt et al., 2019), field campaigns employing multiple platforms were conducted in the Arctic during summer 2014, spring 2015, and summer 2016 to increase our knowledge of Arctic aerosol sources, sinks, chemical transformations and interactions with clouds. During the 2014 campaign unexpectedly high DMS(g) levels were observed (Abbatt et al., 2019; Ghahremaninezhad et al., 2017 and Mungall et al., 2016) and associated with particle nucleation and

growth (Ghahremaninezhad et al., 2016; Willis et al., 2016). Using the global chemical transport model GEOS-CHEM, Mungall et al. (2016) showed that the high levels of DMS(g) observed in the marine boundary layer of the Canadian Arctic Archipelago during summer 2014 largely originated from local marine sources. In particular, measurements made during NETCARE found melt ponds over first-year sea ice to have DMS concentrations of comparable level to the global oceanic



annual average (Gourdal, et al., 2018; Abbatt et al., 2019). This additional source of DMS can have significant consequences for the Arctic aerosol, given the extensive coverage of melt ponds over first-year ice.

Recent modelling studies have examined the impact of DMS on Arctic aerosols. Marelle et al. (2017) updated the WRF-Chem regional model by adding DMS(g) and reported an improvement of surface sulfate estimates in the Arctic. Mahmood et al. (2019) evaluated the impact of DMS(g) emission on the formation of sulfate aerosol, CCN and cloud radiative forcing in global climate model. Although they did not find a significant increase of sulfate aerosol, they predicted higher nucleation rates, increased sulfate deposition, and an increase of cloud droplet number concentration.

In this study, for the first time, we include DMS(g) in the ECCC's online air quality forecast model, GEM-MACH, in order to investigate , at a regional scale, the role of DMS(g) in the formation and growth of aerosols in the Arctic during summertime. Model simulations were carried out for the month of July and the beginning of August 2014, coinciding with the 2014 NETCARE field campaign in the Canadian Arctic to allow comparison with in situ measurements. In what follows, the implementation of DMS in the GEM-MACH model and the simulation setup are described (Section 2), followed by a brief description of the measurement data used for model evaluation (Section 3). Section 4 presents the study results including 1) model simulated DMS(g) and comparison with observations, 2) DMS(g) source sensitivity tests, and 3) DMS(g) impacts on sulfur chemistry and aerosol growth/formation in the Arctic summer. Summary and conclusions of this study are reported in section 5.

## 2. Model and Simulation Setup

The base model used for this study is the Environment and Climate Change Canada (ECCC) air quality prediction model GEM-MACH (Global Environmental Multi-scale model – Modelling Air quality and CHemistry). It consists of an online tropospheric chemistry module embedded within ECCC's numerical weather forecast model GEM (Côté et al. 1998a, b; Charron et al., 2012). The chemistry module includes a comprehensive representation of air quality processes, such as gas-phase, aqueous-phase, and heterogeneous chemistry and aerosol processes (e.g. Moran et al., 2013; Makar et al., 2015a, b; Gong et al., 2015). Specifically, gas-phase chemistry is represented by a modified ADOM-II mechanism with 47 species and 114 reactions (Lurmann et al., 1986; Stockwell and Lurmann, 1989); inorganic heterogeneous chemistry is parameterized by a modified version of ISORROPIA algorithm of Nenes et al. (1999) as described in detail in Makar et al. (2003); secondary organic aerosol (SOA) formation is parameterized using a two-product, overall or instantaneous aerosol yield formation (Odum et al., 1996; Jiang, 2003; Stroud et al., 2018); aerosol microphysical processes, including nucleation and condensation (sulfate and SOA), hygroscopic growth, coagulation, and dry deposition/sedimentation are parameterized as in Gong et al. (2003); the representation of cloud processing of gases and aerosols includes uptake and activation, aqueous-phase chemistry, and wet removal (Gong et al., 2006, 2015). Aerosol chemical composition is represented by eight components: sulfate, nitrate, ammonium, elemental carbon (EC), primary organic matter (POA), SOA, crustal material (CM) and sea salt; aerosol particles are assumed to be internally mixed. A sectional approach is used for representing aerosol size distribution, with either a 2-bin



(0–2.5 and 2.5–10 µm) or a more detailed 12-bin (between 0.01 and 40.96 µm, logarithmically spaced - 0.01-0.02, 0.02-0.04, 0.04-0.08, 0.08-0.16, 0.16-0.32, 0.32-0.64, 0.64-1.28, 1.28-2.56, 2.56-5.12, 5.12-10.24, 10.24-20.48, 20.48-40.96 µm) configuration. A limited area version of GEM-MACH has been in use as ECCC's operational air quality prediction model since 2009 (Moran et al., 2010). GEM-MACH with various configurations has been used in a number of studies, such as air quality and acid deposition in the Athabasca oil sands region (e.g., Makar et al., 2018, Stroud et al., 2018; Akingunola et al., 2018), feedbacks between air pollution and weather (Makar et al., 2015a,b; Gong et al., 2015), and investigating sources and processes affecting the Arctic atmospheric composition in summertime and assessing the impact of marine shipping emissions in the Canadian Arctic (Gong et al., 2018a,b).

### 2.1. DMS flux and oxidation

The emission of DMS(g) from the ocean is determined by the air-sea gas exchange process. In this study, the sea-to-air flux of DMS is parametrised following Liss and Merlivat (1986) and Jeffery et al. (2010) – the latter study combines a global ocean modelling approach and experimental measurements; more details are available in Johnson (2010):

$$F = -K_w \left( \frac{C_g}{K_H} - C_l \right) \tag{2}$$

where $C_g$ and $C_l$ represent the DMS(g) concentrations in the gas and liquid phases, respectively, $K_H$ is the dimensionless Henry's law constant, and $K_w$ is the transfer velocity:

$$K_w = \left[ \frac{1}{K_H k_a} + \frac{1}{k_w} \right]^{-1} \tag{3}$$

where $k_w$ and $k_a$ are the single-phase transfer velocities for the water side (Elliott, 2009) and the air side (Johnson et al., 2010), respectively, which depend on physical properties such as wind speed and air/sea surface temperatures. DMS(g) emissions are assumed to originate from the open ocean, and areas covered by sea-ice were excluded from DMS(g) flux calculation, i.e., the flux in (2) is multiplied by $(1 - fr_{ice})$, where $fr_{ice}$ is sea-ice fraction at a given model grid.

The ADOM-II mechanism does not include DMS. For this study, a DMS(g) oxidation module was added to the GEM-MACH model to account for the oxidation of DMS(g) by OH (through abstraction and addition reactions) and $NO_3$ radicals and the production of $SO_2$; the reaction mechanism is based on Seinfeld and Pandis (1998) with reaction rates from von Glasow and Crutzen (2004). The base mechanism considers $SO_2$ production from the OH-abstraction and $NO_3$ reactions while the OH-addition reaction mainly leads to the formation of MSA. However, as mentioned in the introduction, the OH-addition pathway may also lead to the formation of $SO_2$ via the MSIA-OH reaction. For example, Chin et al. (1996) considered a 75% yield of $SO_2$ production from the DMS OH-addition reaction. The impact of this additional $SO_2$ production pathway is also examined





in this study. No heterogeneous sink for DMS(g) is included. However, reactions of halogen oxide radicals with DMS(g) in the aqueous phase could be significant (von Glasow and Crutzen, 2004; Hoffmann et al., 2016) and need to be considered in the future studies.

### 2.2. Sea-water DMS(aq)

Two sets of seawater DMS data are used in this study: 1) gridded global monthly climatology of surface seawater DMS concentration, at 1° x 1° resolution (Lana et al., 2011; hereafter referred to as CLIM11), and 2) a new satellite-based sea-surface DMS concentration dataset, at 28 km x 28 km resolution, every 8 days (Galí et al., 2018; hereafter referred to as SAT). CLIM11 was developed based on the global surface ocean DMS(aq) measurements collected, mostly, between 1980 and 2009. The monthly climatology was constructed by using interpolation/extrapolation techniques to project the discrete concentration data onto a first guess field followed by further objective analysis (Lana et al., 2011). CLIM11 has been widely used as input to atmospheric chemistry and climate models (e.g. Breider et al., 2017; Marelle et al., 2017). However, there are large uncertainties in the extrapolated ocean DMS climatology over the Arctic, particularly over the Canadian Polar Shelf and the Baffin Bay area due to the scarcity of measurements (Lana et al., 2011; Abbatt et al., 2019).

SAT, the satellite-based DMS(aq) dataset, was developed by Galí et al., (2018) using a remote sensing algorithm that exploits the nonlinear relationship between sea-surface chlorophyll *a*, DMS(aq), its phytoplanktonic precursor dimethylsulfoniopropioante (DMSPt) and plankton light exposure. The satellite algorithm allows for low- and high-DMSP phytoplankton producers (Galí et al., 2015) and for light enhanced DMS concentration in summer (Simó and Pedrós-Alió, 1999), two major factors controlling global DMS distribution and seasonality (Lizotte et al., 2012; Galí and Simó, 2015). The dataset used here is based on the algorithm coefficients fitted for latitudes higher than 45°N, and further optimised for the Arctic Ocean (Galí et al., *submitted*).

Figure 1 compares the averaged July DMS(aq) concentration over the simulation domain from the two datasets, SAT and CLIM11. The same colour scales are used here to allow direct comparison between the two; however, the maximum DMS(aq) concentrations are around 10 and 60 nmol/L for the CLIM11 and SAT datasets, respectively. Both data-sets show relatively high DMS(aq) concentrations in the region of North Atlantic Ocean southeast of Greenland, Gulf of Alaska and Bering strait areas (~ 10 nmol/L). Despite broad agreement in large scale patterns, the SAT dataset shows much more spatial variability than CLIM11 reflecting the higher resolution of satellite observations. In contrast, the CLIM11 dataset shows more uniform DMS(aq) concentrations due to limited DMS(aq) measurements and coarse resolution (555 km interpolation radius). In addition, the SAT dataset also shows high DMS(aq) areas around Makenzie River Delta, Hudson Bay, Labrador Sea, Lancaster Sound and Gulf of St Lawrence, which are not captured in the CLIM11 dataset. Note that over the central Arctic Ocean DMS(aq) concentrations are not available from the SAT dataset due to the limitation of satellite detection in the presence of sea ice (Fig. 1a). As a result, for the model simulation using the SAT seawater DMS, the regions where DMS(aq) is not available were filled in with DMS(aq) values from CLIM11.



### 2.3. Simulation setup

GEM-MACH version 2 with the 12-bin configuration was used for this study. The model domain, centred over the Canadian Arctic on a rotated latitudinal-longitudinal grid with a horizontal resolution of about 15 km, and the model setup are the same as in Gong et al (2018b). The simulation was carried out for July 2014 in this study. Hourly meteorological fields from the global GEM model were used to pilot GEM-MACH. The meteorology was initialized daily (at 00 UTC) using the Canadian Meteorological Centre's regional objective analyses. Output from a global CTM, MOZART-4 (Emmons et al., 2010) was obtained from http://www.acom.ucar.edu/wrf-chem/mozart.shtml for the chemical initial and lateral boundary conditions, including DMS(g). The anthropogenic and biogenic emissions are as described in Gong et al. (2018b); the North America wildfire emissions for the 2014 fire season, archived at the Canadian Centre for Meteorology and Environment Prediction, were used for the simulation. As discussed above, two DMS(aq) datasets were used for the model simulations. In the case of simulation using CLIM1, constant (temporally) climatology for the month of July is used, while in the case of simulation using SAT, DMS(aq) is updated approximately every 8 days whenever the satellite-derived DMS(aq) is available. Figure S1 shows the satellite-derived DMS(aq) concentrations for the SAT time intervals, every 8 days, during July and August 2014 (July 1st to 3rd, July 4th to 11th, July 12th to 19th, July 20th to 27th, July 28th to August 4st and August 5th to 12th).

### 3.  Observational data

The model simulations are compared with DMS(g) measurements made during the NETCARE 2014 summer field campaign, both on board the Canadian Coast Guard Ship (CCGS) *Amundsen* and the Alfred Wegener Institute's Polar 6 aircraft. Figure 2 shows the Polar 6 flight tracks and the Amundsen cruise track during the NETCARE 2014 summer study. The measurements onboard Polar 6 took place from July 4 to 21, consisting of 11 research flights based from Resolute Bay, Nunavut; the measurements onboard Amundsen took place between July 13 and August 7 as the icebreaker sailed through the eastern Canadian Archipelago (Abbatt et al., 2019). Two independent measurements of DMS(g) were conducted on the Amundsen cruise using 1) a Hewlett Packard 5890 gas chromatograph (GC) fitted with a Sievers Model 355 sulfur chemiluminescence detector (SCD) from July 11th to 24th (hereafter referred to as GC-SCD; detection limit of 7 pptv) and 2) a high-resolution time-of-flight chemical ionization mass spectrometer, the Aerodyne HRToF-CIMS from July 15th to August 7th (hereafter referred to as CIMS; detection limit of 4 pptv). For the first method, DMS(g) was collected on GC inlet liner packed with 170 ± 2 mg of Tenax TA®, at the ship's bridge, around 30m above the sea level. Sampling collection time was 300 ± 5 s with a mass flow controlled at approximately 200 ± 20mL/min. The DMS(g) samples were analyzed in less than 24 hours after collection using GC-SCD. The inlet for CIMS was placed in the tower around 16m above the sea surface. Detailed information regarding collection locations, analysis of samples, and uncertainty in measurements with GC-SCD and CIMS are available in Ghahremaninezhad et al. (2017) and Mungall et al. (2016), respectively.



Atmospheric DMS(g) were collected on 5 out of the 11 Polar 6 research flights with the GC-SCD method similar to that used
on board the *Amundsen*. Two Teflon valves were placed before and after the Tenax tube to control the sampling period (300
± 5 s) on the aircraft, and Teflon tubing was used to transfer the sample from outside to the sampler. The samples were stored
in an insulated container with a freezer pack after collection and in a freezer after the flight (more information regarding the
method is available in Ghahremaninezhad et al., 2017).

## 4. Results and Discussions

### 4.1. Simulated DMS(g) and comparison with observations

The modelled monthly averaged DMS(g) mixing ratios for July 2014, at the lowest model level (~ 20 m), using SAT and
CLIM11 DMS(aq) datasets, are shown in Figure 3. Broadly speaking, the two model simulations show similar geographical
distributions of atmospheric DMS over the model domain; generally higher concentration on the western side (Bering Sea and
Bering Strait), up to 900 pptv, and lower on the eastern side, e.g. 50–200 pptv over Baffin Bay, Labrador Sea, and North
Atlantic. This general distribution pattern is consistent with the findings of Sharma et al. (1999) from measurements taken on
an expedition circumnavigating North America including an Arctic Ocean transect in summer/fall 1994; they observed the
highest DMS(g) concentrations over the open waters of the Bering Sea south of the ice edge on the west side of the Arctic
Ocean, up to 50 nmol m$^{-3}$ (or ~1000 pptv), while lower concentrations were observed on the Atlantic side, e.g., 5–10 nmol m$^{-3}$
(or ~100–200 pptv over Labrador Sea). The two simulations result in comparable atmospheric DMS overall. For example,
the DMS(g) averaged over all ocean grids (north of 60 N) for the month of July is 131 pptv using SAT dataset and 145 pptv
using CLIM11 dataset. However, the two simulations do differ on a local scale, e.g., higher DMS(g) mean mixing ratio values
are evident in the figure using the SAT DMS(aq) dataset for some regions such as Hudson Bay (up to 600 pptv using SAT,
while up to 75 pptv using CLIM11) and the Canadian Arctic Archipelago (up to 200 pptv using SAT, compared to up to 100
pptv using CLIM11). The differences in modelled July-averaged DMS(g) between the two simulations, shown in Figure 3(c)
at the lowest model level, largely reflects the differences between the two DMS(aq) datasets.

Figure 4 shows the modelled averaged DMS sea-air flux for July 2014, using SAT and CLIM11 DMS(aq) datasets. The
differences of the two SAT and CLIM11 DMS(aq) datasets (Fig. 1a and 1b) are reflected in the flux values (Fig 4a and 4b);
for example, CLIM11 flux values have less spatial variability due to lower resolution, and SAT DMS flux values are higher
(e.g. >10 µmol m$^{-2}$ d$^{-1}$) at some locations such as the Makenzie River Delta, Hudson Bay, and the Gulf of St Lawrence.
Sharma et al. (1999) reported DMS flux values between 0.007 and 11.5 µmol m$^{-2}$ d$^{-1}$ over Alaska coast (Bering Sea) to the
central Arctic Ocean (Canada Basin) during July-August 1994; Mungall et al. (2016) reported DMS flux values between 0.02
and 12 µmol m$^{-2}$ d$^{-1}$ over the eastern Canadian Arctic. These flux estimates, based on measurements, are comparable with the
present simulations.





### 4.1.1. Comparison with the DMS(g) measurement aboard the Polar 6

Modelled DMS(g) from the two simulations (CLIM11 and SAT), extracted along the Polar 6 flight path coinciding with aircraft samples, are compared with the measurements in Figure 5. This figure includes the simulated average DMS(g) vertical profiles along the flight path. The GEM-MACH model captures DMS(g) mixing ratios close to the measurements and shows a general decay of DMS(g) mixing ratio with height, indicating the influence from local sources of DMS. Using the same observation data, Ghahremaninezhad et al. (2017) highlighted the role of local sources in the Canadian Arctic in DMS emission during

summer. The observation results indicated a decrease in DMS(g) mixing ratios with altitude up to about 3 km, and the largest mixing ratios were found near the surface above ice-edge and open water, coincident with increased particle concentrations (Burkart et al., 2017; Croft et al., 2016a; Croft et al., 2016b; Ghahremaninezhad et al., 2017). The dominant influence of local sources on DMS(g) observed in the Arctic marine boundary layer during summer is further supported by the source sensitivity tests discussed in Section 4.2 later.

The scatter plot in Figure 6 shows the statistical comparison of the model simulations (SAT and CLIM11) with the observation results. Overall, observation and model results are of similar magnitude, but not correlated. The simulation using SAT is in slightly better agreement with the measurement based on root mean squared error and mean bias values of 27.6 and -4.7 compared to 29.5 and -6.6 for the simulation using CLIM11, also better correlation coefficient (as shown in Fig. 6).

### 4.1.2. Comparison with the DMS(g) measurements onboard the *Amundsen*

Figure 7a compares the time series of modelled DMS(g) mixing ratio values from the two simulations (using SAT and CLIM11) following the *Amundsen* cruise track with the GC-SCD and CIMS measurements. The discrepancy between the two measurement datasets can be attributed to the different sampling locations/heights on board the *Amundsen* (e.g. CIMS's inlet

at 16 m above sea level at the bow vs. GC-SCD's inlet on the bridge at 30 m above sea level) and the different sampling/analysis methods. For example, the lower DMS(g) measured by GC-SCD compared to the CIMS measurement on occasions could be attributed to the vertical gradient in DMS(g). This is particularly the case when DMS(g) was mainly driven by local fluxes, e.g., the July 19–21 episode. More details are available in Mungall et al (2016) and Ghahremaninezhad et al. (2017), respectively.

Relatively high DMS(g) mixing ratios were observed during the biologically productive period of July; two main episodes of high DMS(g) concentrations were observed during July 18 – 21 and 26 – 27 when Amundsen was traversing through Lancaster Sound. The modelled DMS(g) from both simulations generally tracked the observations well but both missed the two high DMS(g) episodes in July, though the simulation with SAT captured the episodes of higher DMS in early August well. Overall the model simulations under-predicted the measurements. A statistical evaluation against the CIMS measurements shows

model mean bias of -126.4 pptv (NMB of -57.7%) for the simulation using CLIM11 and -95.4 pptv (NMB of -43.6%) for the simulation using SAT. The large negative biases are mainly driven by the model under-prediction when the icebreaker was



traversing back and forth along Lancaster Sound. The better model results using SAT during the early part of August correspond to the icebreaker's path through Nares Strait, where SAT DMS(aq), up to 9 nmol L$^{-1}$ (Figure S1), is considerably higher than the CLIM11 DMS(aq), < 3 nmol L$^{-1}$, and in better agreement with the in situ DMS(aq) measurement in the area.

Also shown in Figure 7 (b-d) are the time series of the observed (from the Amundsen's Automatic Voluntary Observing Ships System (AVOS) system available onboard the *Amundsen* at ~23m above the sea surface) and modelled surface wind speed, air and sea temperature. These are the physical parameters affecting the sea-air flux. Overall the model is in good agreement with observations, given the model resolution, which suggests that the main cause for the model to underpredict the high DMS(g) events likely lies in the model's representation of DMS sources.

In the next section, we look into several potential DMS source uncertainties which may contribute to the model underprediction.

### 4.2. Source Sensitivity tests

There is a large uncertainty in constraining seawater DMS(aq) in the Arctic due to very few measurements as in the CLIM11 dataset (Lana et al., 2011). Although the satellite-based estimates have the potential to address this shortcoming, they are also subject to uncertainties in retrieval techniques and algorithms. For example, satellite estimation has limitations on ice covered or partially ice-covered ocean surfaces and suffers from uncertainty in satellite products used as input, chiefly chlorophyll *a*, and from uncertainties inherent to algorithm configuration (Galí et al., 2018).

Another potential source of the discrepancies between measurement and model could be due to the neglect of the DMS(g) emissions from ice-covered surfaces. For instance, melt ponds are potential DMS(g) sources (Gourdal et al., 2018). Mungall et al. (2016) estimated that melt ponds can contribute, on average, 20% of atmospheric DMS over and near ice-covered regions of the Arctic during melt season. Here, we conducted a series of source sensitivity tests based on CLIM11, to examine the effects of the potential uncertainty sources and address the discrepancy between the measurement and model results for July

2014, when both CLIM11- and SAT-based simulations had negative bias.

Three sensitivity tests are discussed here:  1) a "no-Ice" model run where sea ice cover is neglected, 2) a model run with enhanced DMS(aq) in Hudson strait and Hudson Bay, 3) a model run with further enhancements in DMS(aq) making the use of the in situ DMS(aq) measurement from the NETCARE campaign. Table 1 lists the setup for these tests. Fig. 8 shows the modelled DMS(g) along the Amundsen path from the source sensitivity tests compared with the observations as well as the

results from the CLIM11 simulation.

Finally, note that DMS(g) underestimation in GEM-MACH might also arise from uncertainty in  sea-air gas exchange coefficients in partially ice-covered waters (e.g., compare Loose et al., 2014 and van der Loeff et al., 2014), too much advection in the meteorological model, or too fast DMS(g) removal in the chemical transport model. Yet, as discussed next, results shown in Fig. 8c support the good performance of GEM-MACH and suggest that most uncertainty arises from DMS concentration in

surface seawater.



### 4.2.1. no-Ice

This sensitivity test (no-Ice) is conducted to examine the effect of neglecting sea-ice cover in the DMS(g) flux calculation. This is essentially an extreme case for considering the potential contribution from melt ponds, by assuming that the entire ice-covered portion of the Arctic ocean is covered by melt ponds, and the DMS(aq) concentrations in these melt ponds are the same as open water sea surface water DMS(aq) concentration, and the flux exchange is the same as over open water.

As shown in Fig. 8(a), by neglecting the sea-ice cover, there is an enhancement in the model simulated DMS(g) mainly over two periods: July 15 when the icebreaker was sailing along the coast of Baffin Island and July 22–25 when the icebreaker was traversing the eastern end of Lancaster Sound (Fig. 2). In both instances the Amundsen was near the melt pond areas or ice edges. The sea-ice fraction in GEM-MACH (based on analysis) shows an area of sea-ice cover over Baffin Bay on July 15[th] and an ice edge located at the eastern end of Lancaster Sound on 22[nd], 2014 (see Fig. S2). However, there is no enhancement in modelled DMS along the Amundsen path during the periods of July 18–21 and July 26–27 when high DMS episodes were observed with the additional DMS sources from sea ice covered surfaces, indicating that the melt pond sources did not contribute to the two high DMS(g) events observed onboard the Amundsen. Figure 9a shows the impact of neglecting sea-ice cover on modelled July mean DMS(g). The highest increase of DMS(g) mixing ratio is around the Chukchi sea (up to 300 pptv).

### 4.2.2. Hudson Strait and Hudson Bay effect (HS-HB)

This sensitivity test is inspired by the sensitivity study conducted by Mungall et al. (2016) and is based on observation results of Ferland et al. (2011). They reported that the productivity of Hudson Strait water is equal to that for the northern Baffin Bay, while the Hudson Bay and Foxe Basin water is about a quarter as productive as northern Baffin Bay (Ferland et al., 2011). We assumed a linear relationship between productivity and DMS(aq) formation, and set the DMS(aq) concentration values in Hudson Strait to be equal to the ship-based measured DMS(aq) values in the northern Baffin Bay (e.g. 20 nmol/L), and the DMS(aq) in Hudson Bay and Foxe Basin to a quarter of that value. The DMS(aq) value (20 nmol/L) is quite consistent with the SAT DMS(aq) in this region (e.g., Fig. S1 a. July 1[st] to 3[rd], b. July 4[th] to 11[th]; the lower SAT DMS(aq) in this region on July 20 and 29 may in part be due to ice condition or other limitation of satellite-based DMS(aq), e.g., potential problem near marginal ice or near ice edge, sparse temporal coverage, etc).

Mungall et al. (2016) conducted this sensitivity test using the GEOS-Chem model, and concluded that the Hudson Bay system contributed significantly to the high DMS(g) event observed onboard the Amundsen during July 18–19. However, the results from our sensitivity test do not seem to indicate that the Hudson Bay system contributed to high DMS(g) events observed onboard the Amundsen (Figure 8b).



The July mean difference in modelled DMS(g) mixing ratios with and without the HS+HB DMS(aq) enhancement indicates
up to 500 and 250 pptv increase of DMS(g) in Hudson Strait and Hudson Bay, respectively from the enhancement. However,
the impact of the Hudson Strait and Hudson Bay system is rather locally confined during the study period (Fig. 9b), indicating
either short DMS lifetime and/or inefficient transport.

### 4.2.3. Updated DMS(aq) in Baffin Bay and Lancaster Sound (CLIM11+ave-Obs)


For this experiment we further updated the DMS(aq) in Baffin Bay and Lancaster Sound using the in situ measurements of
surface seawater DMS(aq) concentration aboard the Amundsen cruise (Mungall et al., 2016). The sampling area is divided
into three sub regions: Lancaster Sound, northern Baffin Bay-Southern Nares Strait, and central Baffin Bay, and each with
averaged DMS(aq) measurement values of 7.9, 11.0 and 4.5 nmol/L, respectively. These values were used to replace the
CLIM11 DMS(aq) values in the respective regions. It is worth noting that these value are comparable to the SAT DMS(aq)
concentrations shown in Fig. 1. By updating DMS(aq) for the sampling region, GEM-MACH predicts the relatively higher
DMS(g) mixing ratios and captures the elevated DMS(g) event days (July 18–22). These results show the importance of the
local source (Lancaster Sound region, for example) in DMS(g) emissions during July (Fig. 8c). Figure 10 compares the CIMS
DMS(g) measured data on board the Amundsen with GEM-MACH simulations, using CLIM11, SAT and CLIM11+ave-Obs.
The statistical evaluations in this figure indicates a significant improvement in CLIM 11 model-observation comparison with
this update (Fig. 10).
Figure 9c shows the difference of July mean DMS(g) mixing ratios using "CLIM11+ave-Obs" and "CLIM11". The DMS(g)
enhancement is largely limited to the locations with the updated CLIM11 DMS(aq) concentration values. The sensitivity tests
result emphasizes the role of locally emitted DMS(g) into the atmosphere particularly in the marine boundary layer.


### 4.3. Impact of DMS on Sulfur Chemistry

In this section we examine the impact of DMS on sulfur chemistry in the Arctic summer through oxidation, production of $SO_2$
and sulfate aerosols. The discussions are based on the simulation results from the "CLIM11+ave-obs" run during July 2014.


### 4.3.1 DMS oxidation and $SO_2$ production

DMS(g) oxidation depends on the oxidants present and the temperature at which the reactions take place. Figure 11 shows the
modelled DMS(g) average chemical lifetime in the atmosphere (Fig. 11a) and the contributions (%) from each of the three
main reaction pathways to DMS oxidation, abstraction with OH (Fig. 11b), addition with OH (Fig. 11c), and abstraction with
$NO_3$ (Fig. 11d), for July 2014.





The DMS(g) atmospheric chemical lifetime (or e-folding time) shown here is based on the decay of DMS(g) due to OH and NO$_3$ radicals in the atmosphere, and it is mostly less than 1 day in the marine environment below the Arctic circle but much longer in the Arctic (Fig. 11a). In the lower Arctic (< 70º N), the chemical lifetime of DMS ranges from less than 1 day to 5 days, and in the high Arctic (> 80º N), the DMS chemical lifetime is between 5 and 20 days, longest over the central Arctic ocean where the concentrations of atmospheric oxidants are lowest. Over the Canadian Arctic Archipelago this DMS lifetime has a large range, from as short as less than half a day (e.g., over Hudson Bay and Hudson Strait, Davis Strait, and southern Baffin Bay) to > 10 days (e.g., Nares Strait, western Queen Elisabeth Islands). The relatively short lifetime over Hudson Bay and Hudson Strait is consistent with the results discussed earlier under sensitivity studies, where we found that the effect of Hudson Bay DMS is mostly confined locally. The DMS(g) life time for the central Arctic was predicted as 2.5 days by Leck and Persson (1996). Sharma et al. (1999), by using a 1-dimensional (1-D) photochemical box model, estimated that the lifetime of DMS(g) due to OH and NO$_3$ oxidation to be around 6-8 days in the central Arctic during August. The relatively shorter lifetime of DMS(g) from Sharma et al. (1999) compared to our study may be partly due to slightly higher OH concentration simulated by their 1-D model (e.g., 0.01 – 0.02 pptv in their August case versus 0.006–0.01 pptv in this study). In addition to OH and NO$_3$ radicals, halogenated radicals may have an important role as an additional sink for DMS(g) in the Arctic atmosphere (von Glasow et al., 2004; Hoffmann et al., 2016). The multiphase chemistry of DMS(g) and the impacts of the halogens in the DMS(g) chemistry/lifetime/products in the Arctic atmosphere need to be considered for future GEM-MACH model studies.

The NO$_3$ concentrations, and as a result, the DMS(g) oxidation by NO$_3$, decrease sharply above 70º N. On the other hand, the oxidation by the OH radical is more important north of 70º N during the bright month of July. Overall, the abstraction pathway with the NO$_3$ radical (up to 95%) below the 70º N and addition pathway with the OH radical (up to 90%) above 70º N are the dominant oxidation pathways for DMS(g) in the sub-Arctic and Arctic, respectively.

SO$_2$ is one of the important products of DMS(g) oxidation in the atmosphere. SO$_2$ concentrations were altered in the GEM-MACH model by including DMS(g) as a new biogenic source. The SO$_2$ increment (July-averaged) due to DMS (or DMS-derived SO$_2$, both absolute and relative to total modelled SO$_2$) are shown in Figure 12. The absolute SO$_2$ concentration difference in Fig. 12 (upper panel) is up to ~600 pptv, and the relative contribution of DMS-derived SO$_2$ to total SO$_2$ is up to almost 100% for some regions (lower panel), which could be significant for the remote and clean Arctic environment during July. SO$_2$ concentrations are increased in Hudson Bay and south Baffin Bay (around 100 pptv) by adding DMS(g) in the model. The absolute values of DMS-derived SO$_2$ are small in these areas in comparison with other areas but due to the low background SO$_2$ concentrations, DMS makes a significant contribution to SO$_2$ in these areas as shown in Fig. 12b. The relative SO$_2$ increment plots (Fig. 12b) highlight the significant change of SO$_2$ in the atmosphere by including DMS in the model. The SO$_2$ concentrations are relatively low in the Arctic clean atmosphere during summer, and the relative increase of SO$_2$ due to DMS(g) is more than 70% over most of the Arctic Ocean.






### 4.3.2 Sulfate aerosols

The SO$_2$ formed from DMS(g) oxidation will further undergo oxidation in the atmosphere by OH radical to form sulfuric acid, which can either nucleate to form new particles or condense on existing particles. In GEM-MACH, the nucleation and

condensation of sulfuric acid are treated as two competing processes. The H$_2$SO$_4$-H$_2$O nucleation rate is parameterized following Kulmala et al. (1998), and the condensation rate is parameterized based on the modified Fuchs-Sutugin equation (Fuchs and Sutugin, 1971). The combined nucleation-condensation equation is solved using an accelerated iterative scheme as described in Gong et al. (2003). The sulphate mass produced by nucleation is placed in the model's smallest size bin; the treatment of condensational growth of particles is handled by the same mechanism as described in Jacobson et al. (1994). As

a result, the inclusion of DMS will induce changes in modelled aerosols in GEM-MACH, both in mass concentration and size distribution (Croft et al., 2016a).

Figure 13 shows the changes (both absolute and relative) in modelled July-averaged aerosol sulfate mass concentration due to DMS (or the production of biogenic sulfate) at the lowest model level. This difference is in the range of 1 to 20 ng/kg (< 10%) in the high Arctic (< 80° N) and is higher in the lower Arctic (e.g. up to 100 ng/kg in Baffin Bay). Also, the increase of sulfate

mass is significant for the east and southwest of the domain with higher DMS(g) (e.g., North Atlantic and off the coast of southern Alaska).

Figure 14 shows the relative mass change due to DMS chemistry in aerosol sulfate (July-average) for four different size ranges: 10–50, 50–100, 100–200, and 200–1000 nm. Examination of the different size ranges indicates that the most significant relative sulfate additions due to DMS reside in the smaller sizes (10–200 nm). It is interesting to note that at higher latitudes (> 70° N)

the enhancement in sulfate due to DMS is more pronounced in the size range of 50 to 100 nm. This is in contrast to the enhancement at the lower latitudes (< 70° N) which is more evident in the size range of 200–1000 nm. This could be an indication for more favourable conditions for nucleation under the cleaner environment at high latitudes while condensation onto existing aerosol is favoured for DMS derived sulfuric acid at lower latitudes, as found in Leaitch et al. (2013).

Abbatt et al. (2019) showed the highest increase in particle number concentration to be between 15 and 50 nm at Alert, Nunavut

during July and August associated with new particle formation and growth from natural sources (see Fig. 7 of Abbatt et al., 2019). They estimated the contribution of natural sources to particles in 30 to 50 nm size range to be around 20% to 70%. Figure 14 shows 20-50 % and ~ 50% increase of sulfate particle mass between 10 to 50 nm and 50 to 100 nm, respectively, for July around Alert. In general, GEM-MACH suggests the enhancement of particles between 50 to 100 nm to be higher than particles between 10 to 50 nm for the high Arctic. This difference between Abbatt et al., (2019) and GEM-MACH results could

be partly due to missing other natural sources (e.g., organics, see Burkart et al., 2017; Willis et al, 2016) in the model. Possible inadequacy in model representation of particle nucleation process may also contribute to the size discrepancy between model and observation. For example, in the model new particles formed through nucleation are added to the first model size bin (10 – 20 nm), at sizes considerably bigger than nucleating particles in the real world (e.g., Kulmala et al., 2006).



The modelled size-resolved sulfate increments due to DMS are compared to the measurements of size-resolved biogenic sulfate

onboard the Amundsen cruise during the 2014 NETCARE campaign. Size segregated aerosol samples were collected and analyzed for sulfur isotopes in order to apportion total aerosol sulfate into different origins, biogenic, anthropogenic and sea salt (Ghahremaninezhad et al., 2016). The modeled size-resolved aerosols were mapped onto the size ranges of the observation. Figure 15 shows the comparison for 3 smaller size ranges: < 0.49 µm, 0.49–0.95 µm, and 0.95–1.5 µm. The aerodynamic diameters were converted to dry diameters by using a dry diameter correction factor of 2.3 (Ming and Russell, 2001). Based

on their analysis, Ghahremaninezhad et al. (2016) indicated that most of the biogenic sulfate resides in the smallest size range (< 0.49 µm) (refer to Figure 6c in Ghahremaninezhad et al., 2016). The model simulation in this study compares well with the observations and also demonstrates that a larger fraction of DMS derived sulfate (or biogenic sulfate) is found in aerosols with sizes < 0.49 µm, particularly at higher latitudes.

During July 2014, Leaitch et al. (2016) found a strong influence of the particles in the size range of 20 to 100 nm in the cloud

droplet number concentrations (CDNC) in liquid clouds in the eastern Arctic over Resolute Bay area. Quinn et al. (2017) also emphasised the role of the non-sea-salt sulfate aerosol as the primary CCNs between 70º S and 80º N. Also, Ghahremaninezhad et al., (2016) found that biogenic sulfate particles are the dominant non-sea-salt particles during July in the Arctic atmosphere, and > 63 % of non-sea-salt fine aerosol (<0.49 µm) were from biogenic source (DMS). Our modeling results indicate the formation/growth of biogenic sulfate aerosol in the size range of 10-200 nm. These results suggest that the non-sea-salt sulfate

aerosol in the summertime Arctic is dominated by fine and ultrafine biogenic particles, which may act as CCNs and/or influence CDNCs and play a climatic role.

### 4.3.3    Impact of possible $SO_2$ formation from the OH-addition pathway

We examined the impact of possible additional $SO_2$ formation from the OH-addition pathway via the MSIA-OH reaction, which is the "crossover point" between the addition and abstraction pathways (von Glasow and Crutzen, 2004). In this oxidation sensitivity test we considered a 75% yield of $SO_2$ from the OH-addition following Chin et al. (1996). Figure 16 shows the relative percentage of the July averaged $SO_2$ with and without 75% yield of $SO_2$ from the OH-addition pathway. In this figure the difference (up to > 50% over the central Arctic, and 30-50% over the Canadian Archipelago) is more pronounced

above 70º N, where the effect of OH addition pathway is significant. Since OH-addition dominates DMS oxidation in the Arctic environment (particularly the high Arctic) due to low temperature (as discussed above), this additional $SO_2$ formation mechanism can be important here.

Figure 17 shows relative (percentage) change in the modelled averaged sulfate aerosol with and without 75% yield of $SO_2$ from the OH-addition pathway for different size ranges. This pathway exerts an important influence on the fine sulfate aerosol

with the sizes of the 10-200 nm (mostly 50-100 nm) above 70º N. As shown in figure 17, the increase of sulfate aerosol mass in the size range 10–200 nm is more than 30% for some regions, due to adding this pathway. These results emphasise the potential importance of the "crossover point" between the addition and abstraction pathways above 70º N.



Leaitch et al. (2013) estimated the July and August averaged sulfate mass concentrations at Alert to be equal to 25 ng/m$^3$ and
84 ng/m$^3$ for the particles in the size ranges of 20–100 nm and 20–200 nm respectively (based on a 100% conversion from
integrated volume concentration measured under clean condition). However, assuming that sulphate comprises about 40% of
the submicron particles in the clean air at Alert (Leaitch et al., 2018), the sulphate mass concentrations are estimated in the
range of, at least, 10-35 ng/m3 at Alert from natural sources. The GEM-MACH results show increase in July averaged sulfate
particle mass due to DMS to be slightly less than 10 ng/m$^3$ without 75% yield of SO$_2$ and more than 15 ng/m$^3$ with 75% yield
of SO$_2$ in the size range of 10-200 nm at Alert.

## 5. Conclusion

In this study, we implemented a DMS representation in the GEM-MACH model for the Arctic domain. Two sets of seawater
DMS(aq) data, CLIM11 and SAT, were used as the source of the atmospheric DMS(g).
We compared the GEM-MACH simulation results with the DMS(g) measurements aboard the Polar 6 aircraft and on board
the Amundsen from the NETCARE field studies on July 2014. Overall, the modelled DMS(g) from both CLIM11 and SAT
simulations tracked the observations, however, both underpredict the two high DMS(g) concentrations events in July. To
consider the discrepancy between the measurement and model results, we conducted source sensitivity studies by using the
CLIM11 dataset. GEM-MACH represents better agreement with the measurement by adjusting the CLIM11 DMS(aq) dataset
and using measured average DMS(aq) concentration values over Baffin Bay and Lancaster Sound area. In general, the
dominant influence of local sources on DMS(g) observed in the Arctic marine boundary layer during summer is supported by
the conducted source sensitivity tests. The CLIM11 climatology clearly does not reflect the marine source well in the Arctic
due to very limited observations available. The satellite-derived sea surface water DMS dataset has the potential to address
this shortcoming as it seems to better reflect the high spatial and temporal inhomogeneity in DMS(aq) production. However,
further development in retrieval algorithms is needed in addressing some of the limitations in the Arctic environment (IOCCG,
2015), e.g., over partially ice-covered sea surfaces (Bélanger et al., 2007), and in coastal waters with high loadings of
continental materials (Mustapha et al., 2012).
Also, for this first implementation of DMS in the GEM-MACH model, the DMS(g) oxidation occurs in the addition and
abstraction pathways with the main oxidants, OH and NO$_3$ radicals. The simulation results show that the abstraction pathway
with the NO$_3$ radical (up to 95%) below 70º N and addition pathway with the OH radical (up to 90%) above 70º N are the
dominant oxidation pathways for DMS(g) in the Arctic and sub-Arctic. Neither aqueous phase oxidation nor halogen chemistry
were included in this study. Both can be important additional DMS oxidation pathways and further studies are needed to
determine their role in the Arctic environment.
By adding DMS(g) in the GEM-MACH model, the atmospheric SO$_2$ concentration increased (up to ~100% for some regions).
This increase in may play a significant role in the growth and nucleation of aerosols. The enhancement of sulfate biogenic



aerosols was also more pronounced in the size range 10–200 nm. These fine-ultrafine particles are able to affect the climate indirectly by altering the CDNCs (e.g., Leaitch et al. 2016).

In addition, the role of $SO_2$ formation from the OH-addition pathway via the MSIA-OH reaction was examined in the GEM-MACH model. Results indicated the importance of this pathway in the formation of $SO_2$ and sulfate aerosol above 70º N.

This study highlights the importance of DMS(g) in the formation and growth of aerosols in remote areas, such as the Arctic atmosphere during summer. More broadly, our results stress the need for adding interactive marine DMS emission and subsequent atmospheric processes (including oxidants) in IPCC-class climate models, if we are to resolve ocean-atmosphere feedbacks in the changing Arctic environment and globally (Charlson et al., 1987; Levasseur, 2013). Further investigations and measurements are necessary to see the impact of DMS(g) in the $SO_2$ concentration and sulfate aerosol distributions.


*Data availability*. Data are available by email request (roghayeh.ghahremaninezhad@canada.ca).

*Author contributions.* WG and RG designed the study. RG implemented DMS in the model with the supervision of WG and assistance from SRB, AA, QZ and AL. MG and Martine L provided the satellite and measurement DMS(aq) data with the
supervision of Maurice L. RG performed the measurement by supervision of ALN and WRL. RG prepared the manuscript with contributions from all co-authors.

*Acknowledgements.* We are thankful for the contribution of the Air Quality Research Division of the Environment and Climate Change of Canada in this study, especially, Kenjiro Toyota, Balbir Pabla and Verica Savić-Jovčić. The measurement studies
were part of the NETCARE project, and the authors would also like to thank the principal investigator J. P. D. Abbatt, the crew of the *Amundsen*, Polar 6 and fellow scientists.

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





**Table 1: DMS(aq) data-sets and sensitivity runs.**

| Name | Explanation |
|------|-------------|
| CLIM11 | DMS(aq) based on climatology (Lana et al. 2011) |
| SAT | DMS(aq) based on satellite observation (Galí et al. 2018) |
| no-Ice | DMS(aq) using CLIM11; the ice coverage is ignored |
| HS+HB | DMS(aq) by updating CLIM11 for Hudson Bay/Strait and Foxe Basin |
| CLIM11+ave-Obs | DMS(aq) using HS+HB with further updating the CLIM11 with the in-situ measurement data over Lancaster Sound, northern Baffin Bay-Southern Nares Strait, and central Baffin Bay |



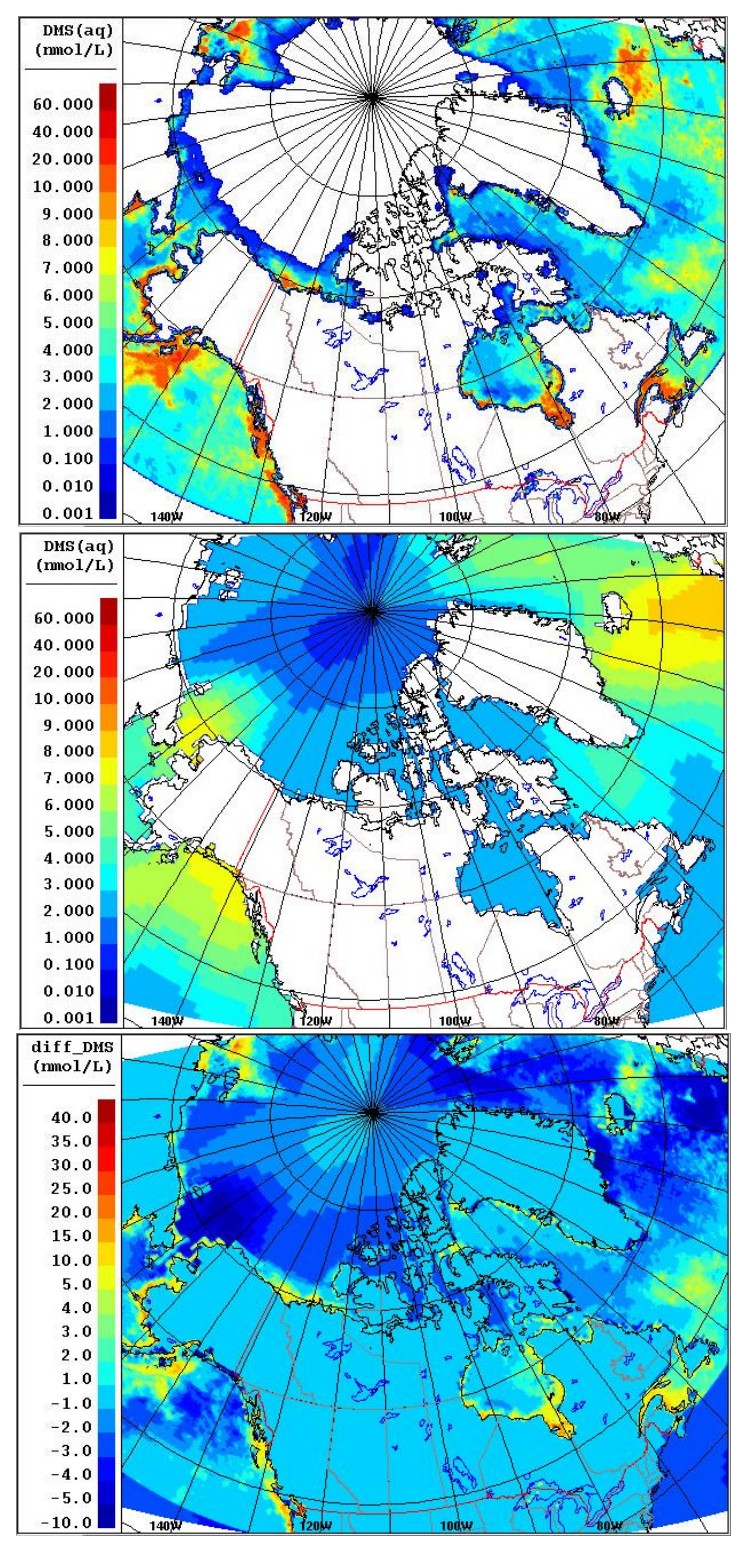






**Figure 1: July average of the DMS(aq) concentrations (nmol/L), SAT (upper panel) and CLIM11 (middle panel) and the absolute difference of DMS(aq) SAT-CLIM11 (lower panel).**











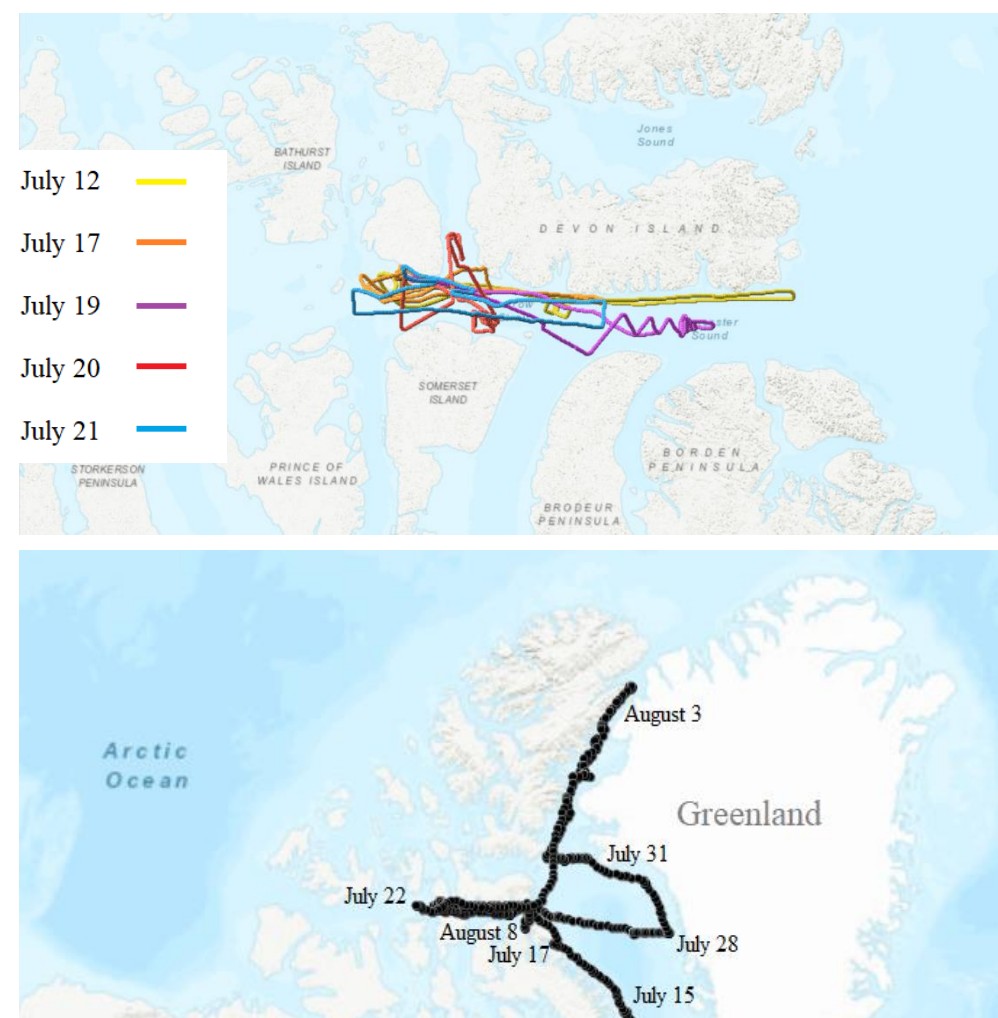

**Figure 2: The Polar 6 (upper panel) and Amundsen (lower panel) trajectories during July 2014.**










**Figure 3: DMS(g) mean mixing ratios for July 2014, at 1.5 m, by using SAT (upper panel) and CLIM11 (middle panel). The lower panel shows the difference of SAT and CLIM11 DMS(g).**










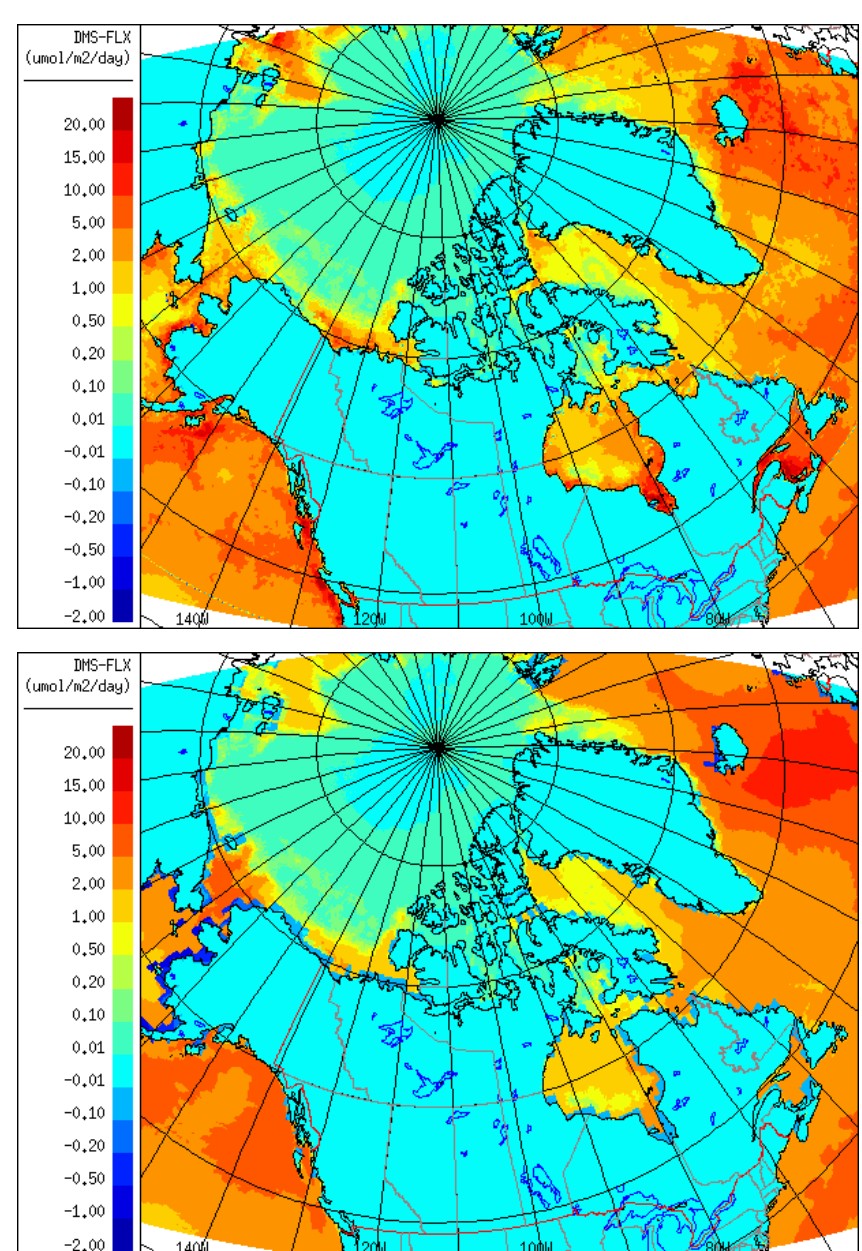

**Figure 4: DMS sea-air average flux for July 2014, at 1.5 m, by using SAT (upper panel) and CLIM11 (lower panel).**





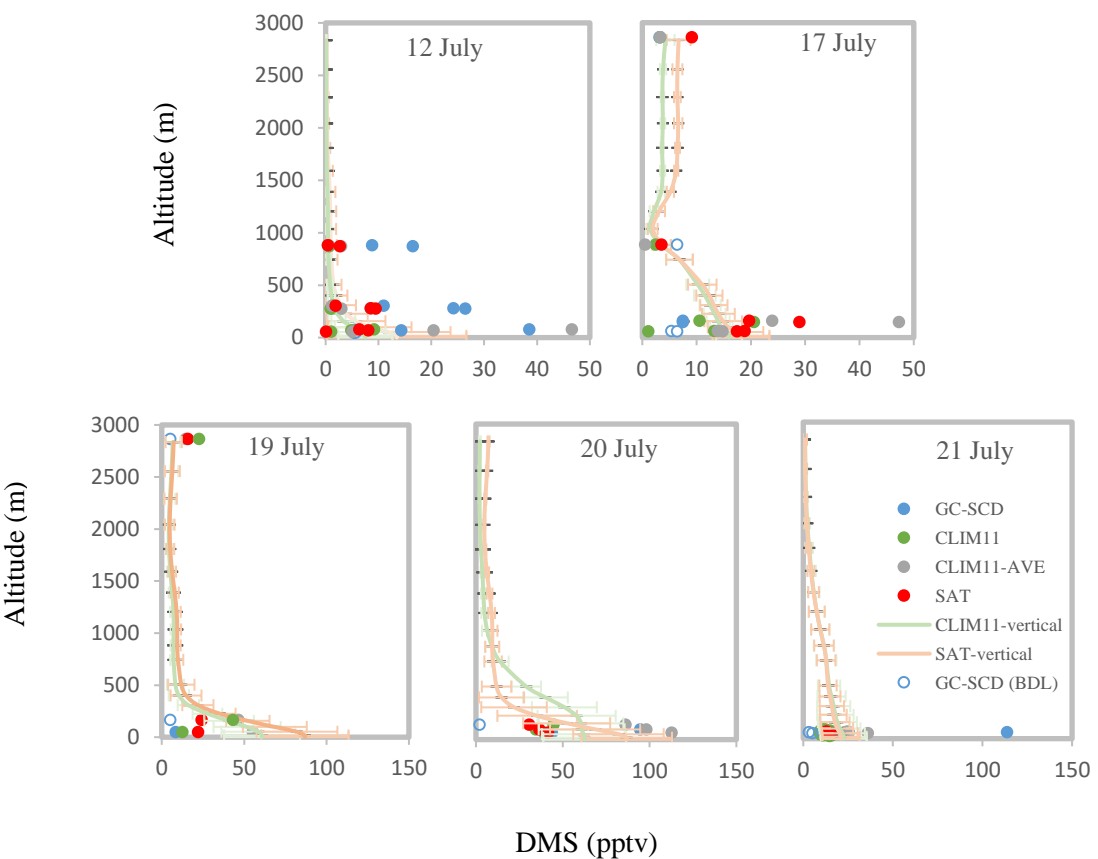

DMS (pptv)

**Figure 5: Dots: DMS(g) from the GEM-MACH simulations (CLIM11, CLIM11+ave-obs and SAT) extracted along the Polar 6 flight path coinciding with observation data (GC-SCD) during July 2014. The measurement values below the detection limit (BDL) are shown with empty circles. Lines: the simulated average DMS(g) vertical profiles during the flights using SAT and CLIM11 datasets. The error bars indicate the standard deviations.**




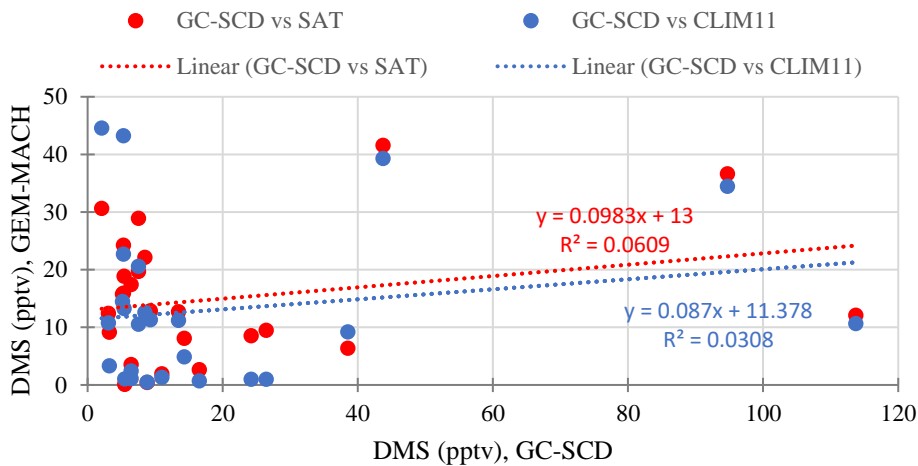

**Figure 6: The GC-SCD observation results aboard the Polar6 aircraft versus model simulations (SAT and CLIM11) during July 2014.**












**Figure 7: a. Campaign-mean DMS(g) mixing ratios for the co-sampled GEM-MACH simulations (Sat and Clim11) and measurements on board the Amundsen from July 11ᵗʰ to 24ᵗʰ (GC-SCD), and from July 15ᵗʰ to end of July (CIMS). b-d. Wind speed and sea-air temperatures from the *Amundsen*'s AVOS system and GEM-MACH.**











**Figure 8: Sensitivity runs during July 2014: a. no-Ice, b. HB+HS, c. CLIM11+ave-Obs.**







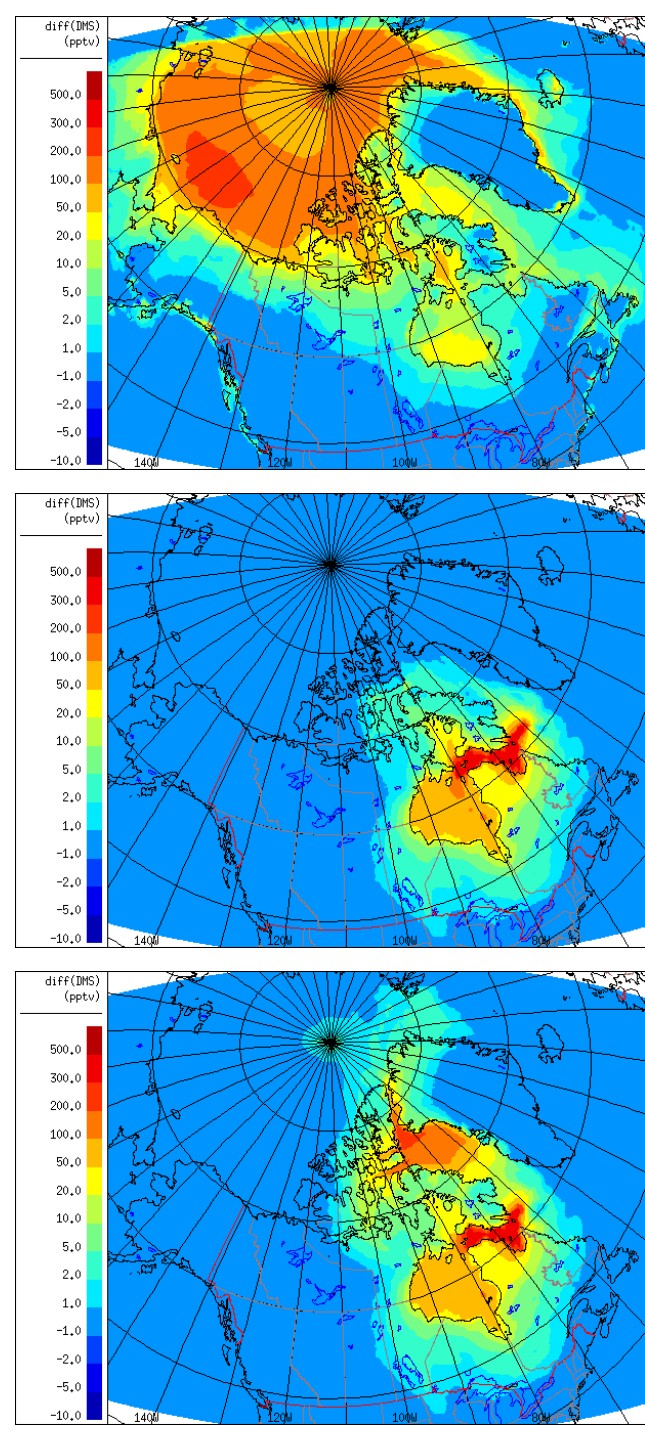

**Figure 9: The difference between DMS(g) July average from Lana with a. no-Ice, b. HB+HS, c. CLIM11+ave-Obs.**





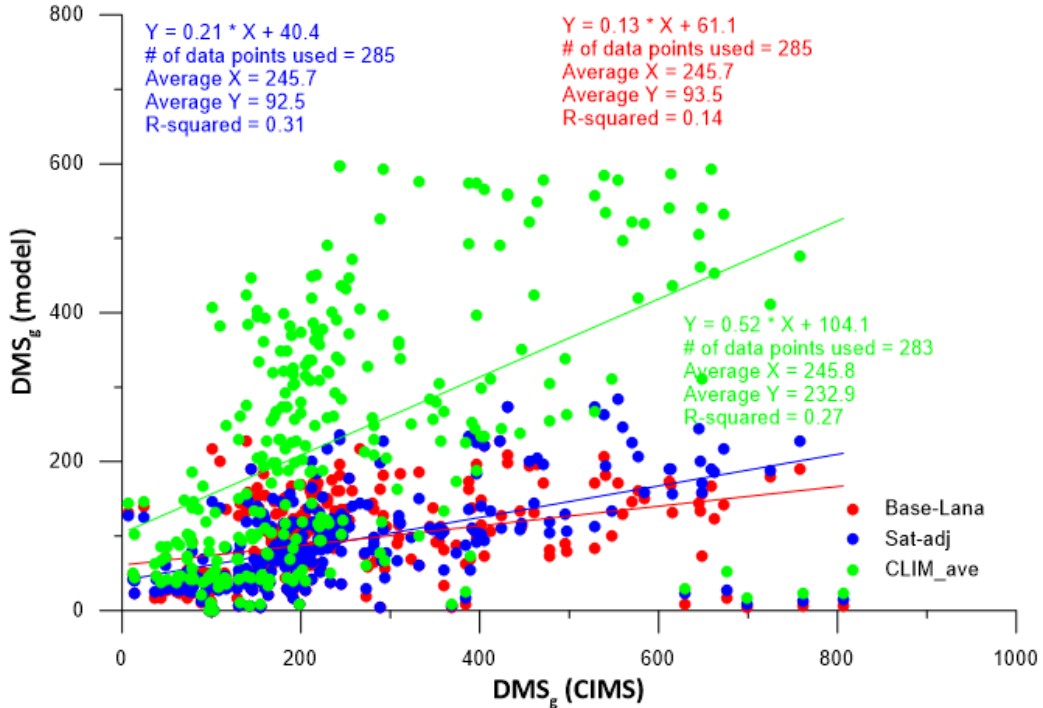

**Figure 10: The comparison of DMS(g) measured by CIMS on board the Amundsen (July-August 2014) with GEM-MACH simulations, using CLIM11 (red), SAT (blue) and CLIM11+ave-Obs (green).**





**Figure 11: a) DMS(g) average lifetime for July 2014, and the percentage (%) of the each pathway; b) abstraction with OH, c) addition with OH, and d) abstraction with NO₃.**





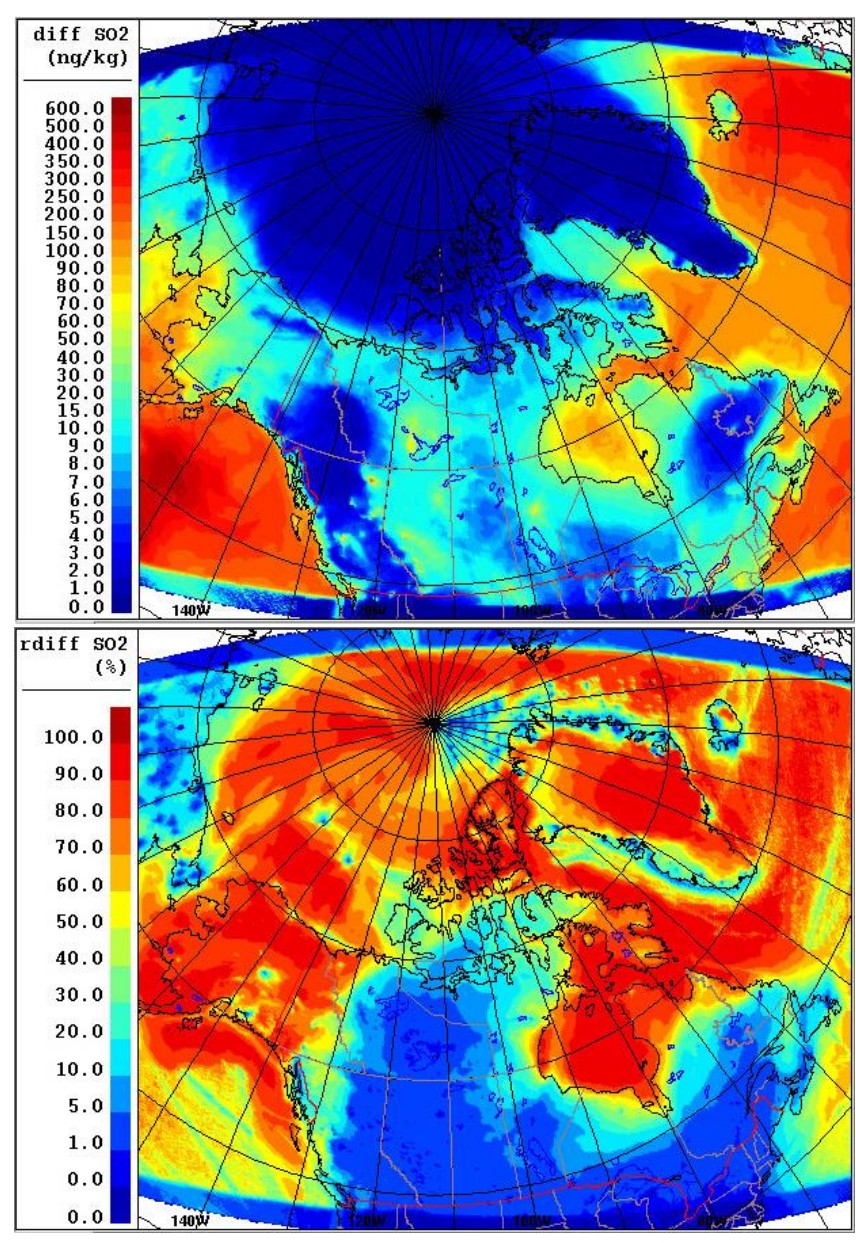

**Figure 12: The absolute (upper panel) and relative (lower panel) SO₂ increment during July 2014 due to DMS in GEM-MACH.**



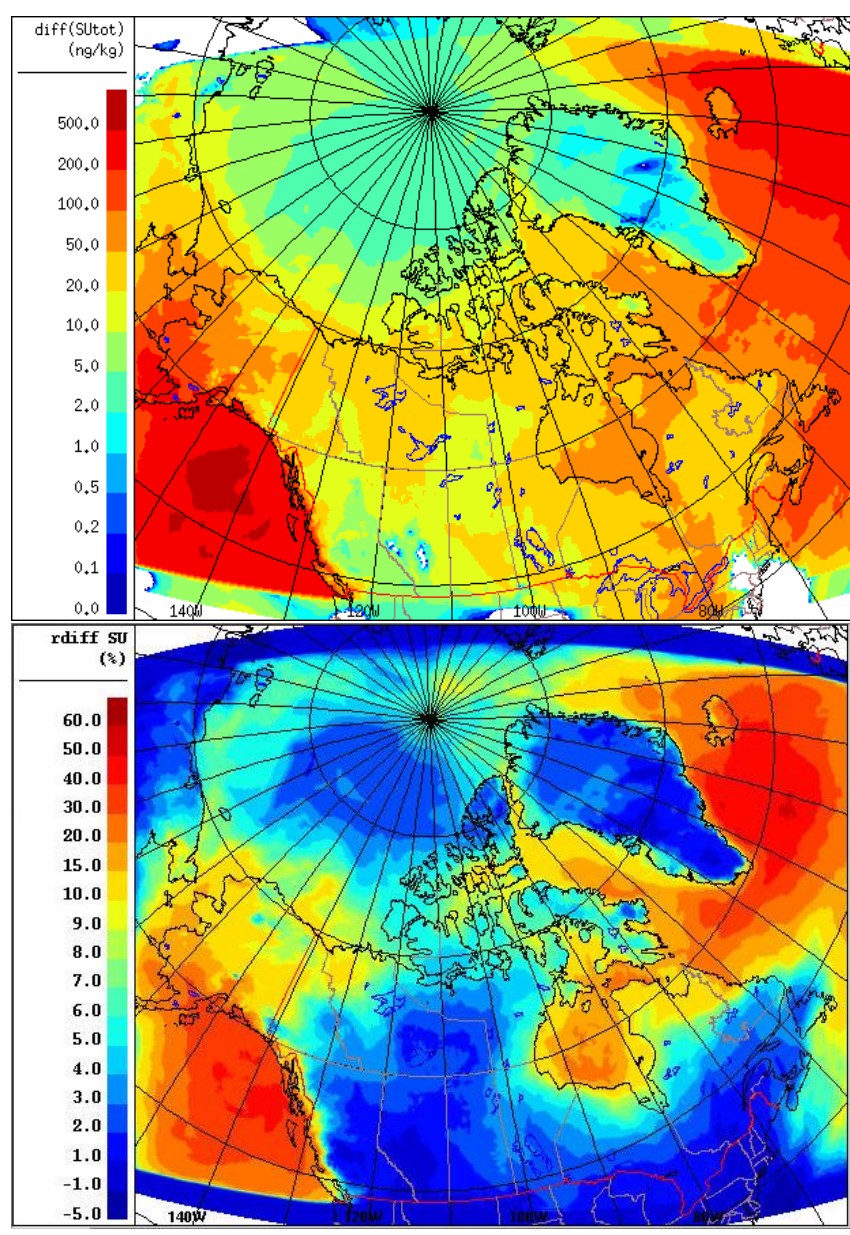


**Figure 13: The absolute (upper panel) and relative (lower panel) differences of the non-sea-salt sulfate aerosol concentrations (0.01 to 41 μm) with and without DMS(g) during July 2014.**






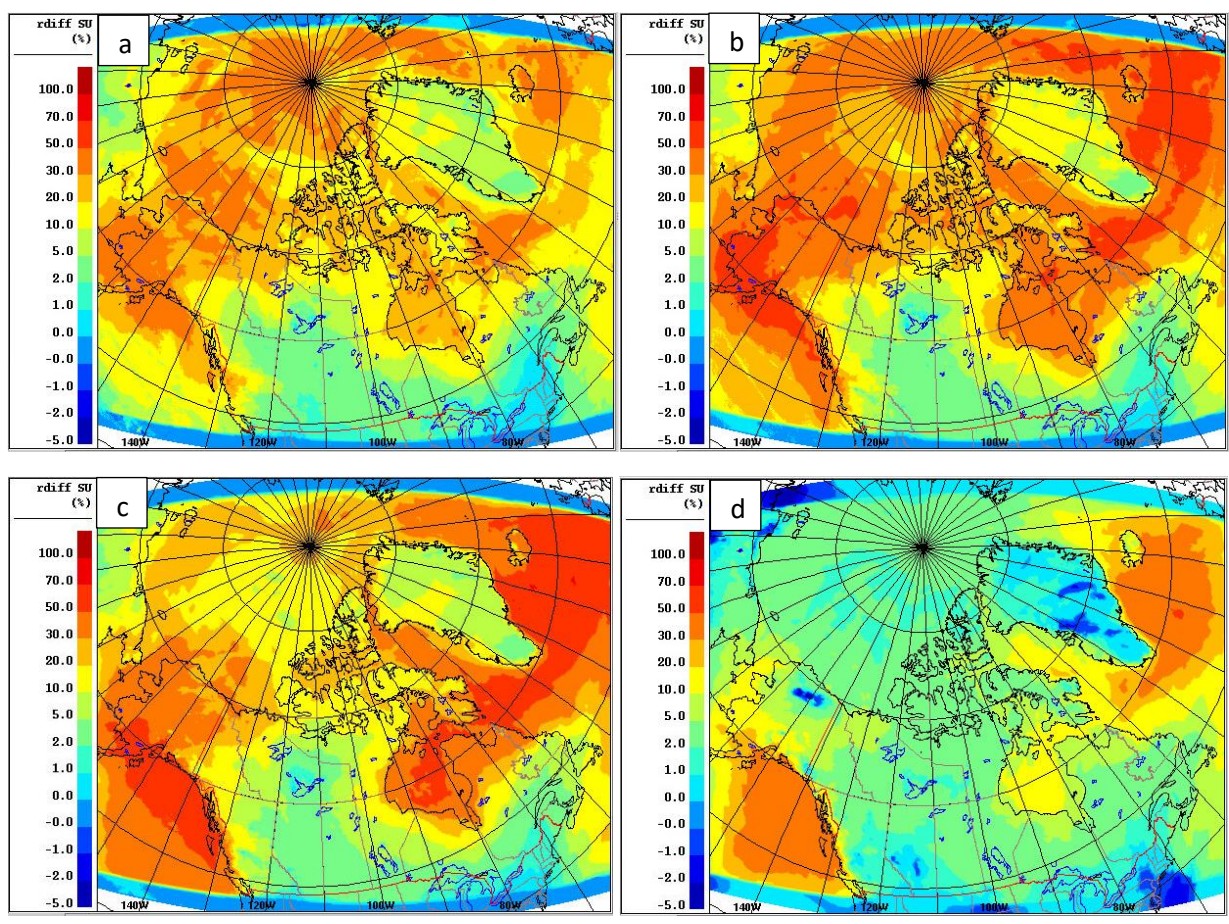

**Figure 14: The relative mass change of aerosol a) 10 – 50 nm, b) 50 – 100 nm, c) 100 – 200 nm, and d) 200nm – 1 μm, with and without DMS(g) during July 2014.**








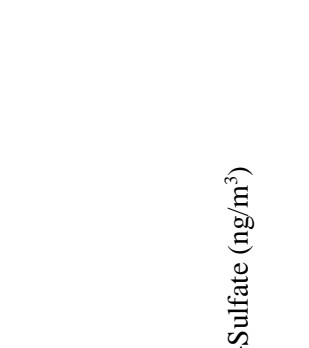

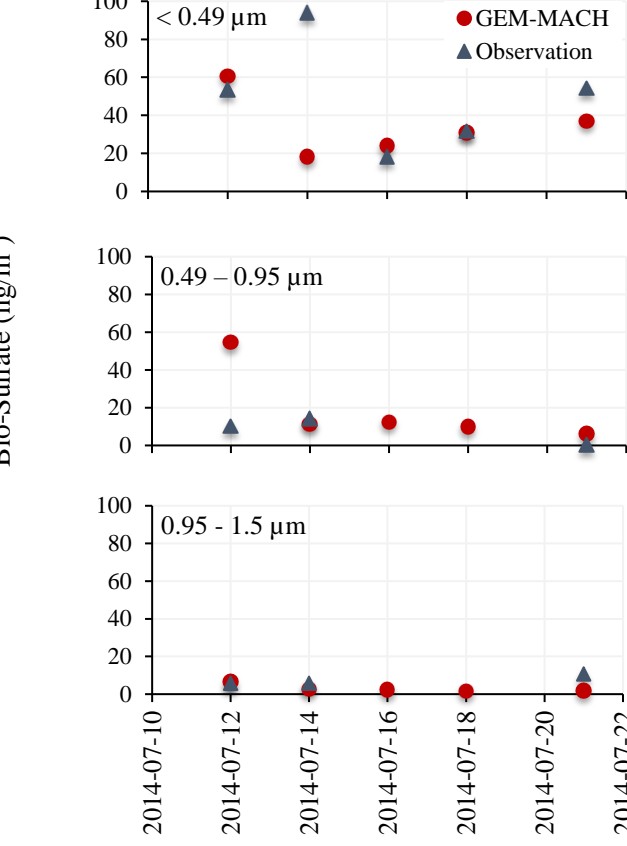

**Figure 15: Measured and simulated 2-day average biogenic fine sulfate aerosol (0.49 µm, 0.49 – 0.95 µm, and 0.95 – 1.5 µm) during July 2014.**







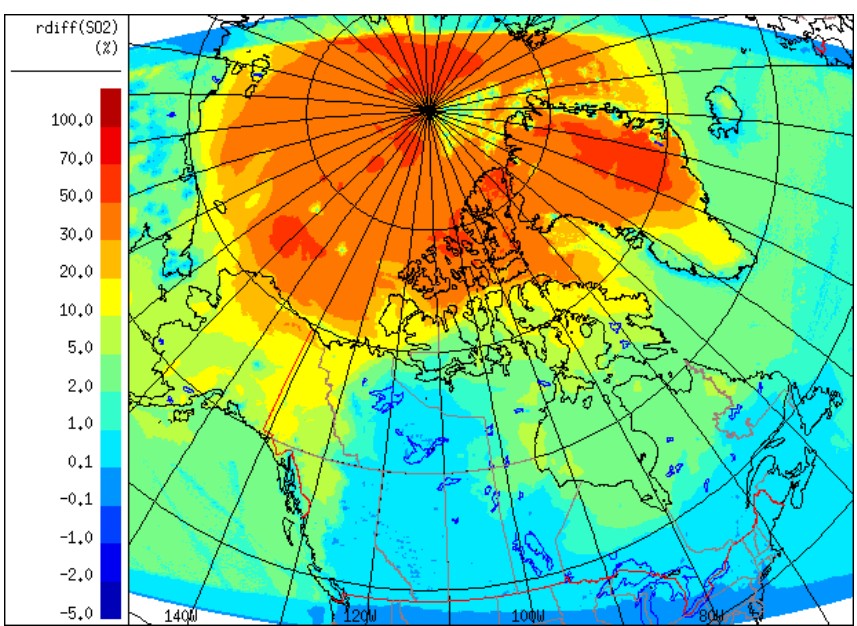

**Figure 16: The relative percentage of the July 2014 averaged SO₂ with and without 75% yield of SO₂.**





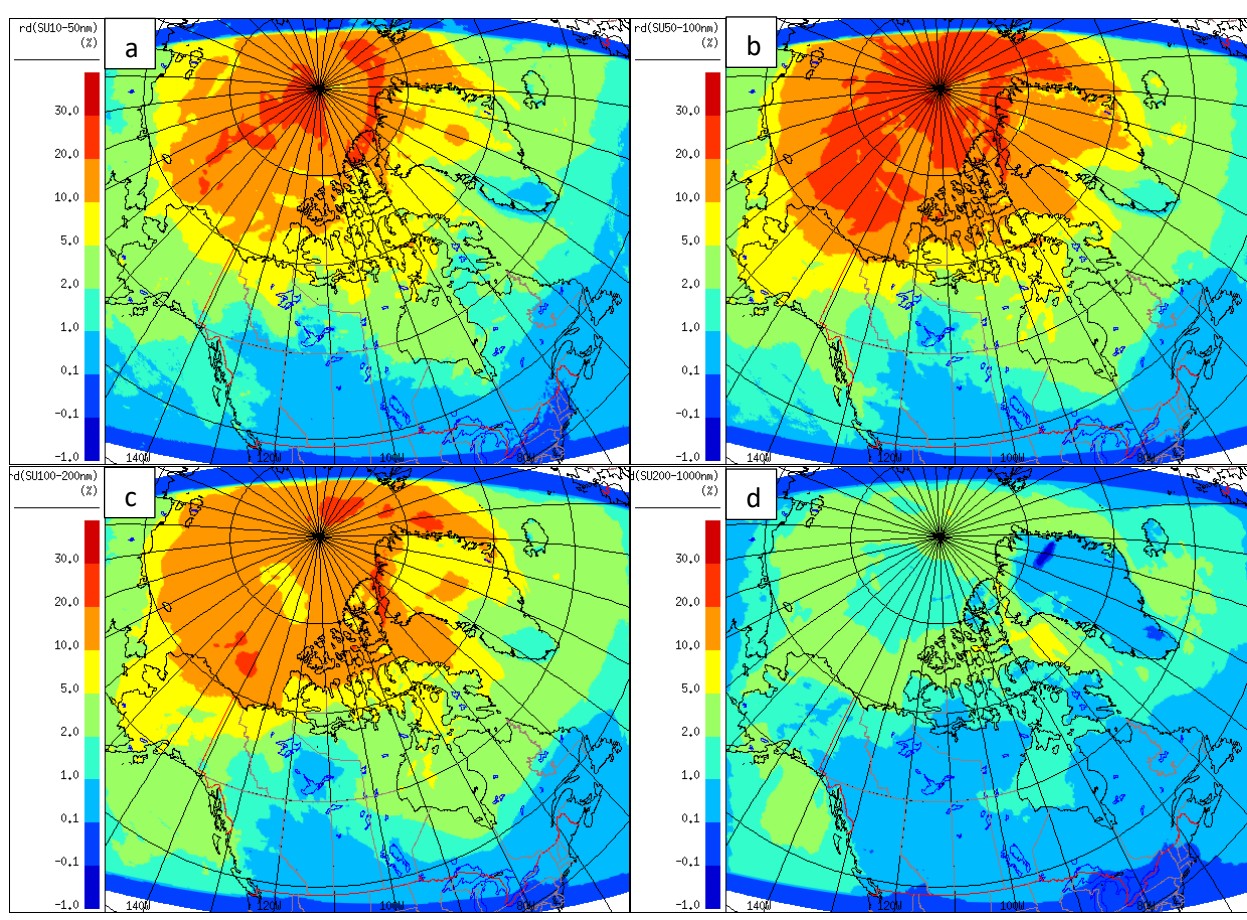

**Figure 17: The relative difference of aerosol a) 10 – 50 nm, b) 50 – 100 nm, c) 100 – 200 nm, and d) 200nm – 1 µm, with and without 75% yield of SO₂ from the OH-addition pathway for July 2014.**