# Peer review of "Dimethyl sulfide and its role in aerosol formation and growth in the Arctic summer – a modelling study"

_Atmospheric Chemistry and Physics, 2019_

## Referee Comment (RC1) · Anonymous Referee #1 · 26 Jun 2019

General comments

The paper presents a modelling study on DMS in the atmosphere and the role of the DMS-derived sulfate in aerosol formation and growth. The model is fed by satellite experimental data and a comparison between model results and experimental data is carefully analyzed and clearly presented. It's a pity that the formation of MSA from DMS is only mentioned at 155-160 and not considered in the model GEM-MACH. MSA in the Arctic presents lower concentration than sulfate but it is very efficient in new particle formation and growth of existing particles, for this reason it constitute the main uncertainty in the modellization of biogenic aerosol formation. Anyway, the paper is

focused on biogenic sulfate, results from models are compared with sulfate measurements therefore, approach, and results are correct. The paper is of high quality and surely deserve the publication on ACP with just few minor revisions.

Specific comments

Lines 40-42. Among the sources of sulfate, the volcanic is not reported here. In north hemisphere, and especially in Iceland, the presence of volcanoes and fumaroles can furnish a contribution to SO2 budget.

Line 207. CLIM11 instead of CLIM1.

Lines 229-232. Here the sampling height (of mean height) of sampling has to be reported. As reported below in the paper, this information is useful as the DMS concentration depend to the height of sampling.

Lines 364-365. This sentence is not clear, what is clear from figure 8b is that the difference between DMS results from CLIM11 with and without HS+HB DMS(aq) are very low.

Lines 460-463. The size discrepancy between model and observation could be due to the effect of MSA in nucleation processes that is not considered by the model. A sentence or a very short discussion on this effect could be add here.

---

## Author Comment (AC1) · 6 Jul 2019

We are thankful for the referee#1's great comments. Please see below: a point-to-point response to the comments (in *italic font*).

1. Lines 40-42. Among the sources of sulfate, the volcanic is not reported here. In north hemisphere, and especially in Iceland, the presence of volcanoes and fumaroles can furnish a contribution to SO2 budget.

   *Yes, volcanoes and fumaroles do contribute to global atmospheric sulfur budget. It is believed that they contribute to 5 – 7 % of global total sulfur emissions. In the manuscript, we had mentioned the main sources during the Arctic summer. However, we have now added the geological sources as well, to make a general statement. Here are the modifications to address this comment:*
   *(Line 41-42): Sulfate aerosols in the Arctic atmosphere originate from the anthropogenic, sea salt, geological and biogenic sources*
   *(Lines 45-46): The geological sources include $SO_2$ emission such as from volcanoes and Smoking Hills (Yang et al., 2018; Rempillo et al., 2011).*

2. Line 207. CLIM11 instead of CLIM1

   *Thanks, we fixed that.*

3. Lines 229-232. Here the sampling height (of mean height) of sampling has to be reported. As reported below in the paper, this information is useful as the DMS concentration depend to the height of sampling.

   *Great point. We agree that the sampling heights need to be mentioned, since DMS concentrations change with the altitude. We added the sampling heights: (Line 230): 50 to 5000 m above the mean sea level.*
   *Please note, as mentioned in the manuscript, more details about the observation methods/sampling heights are reported in Ghahremaninezhad et al., 2017.*

4. Lines 364-365. This sentence is not clear, what is clear from figure 8b is that the difference between DMS results from CLIM11 with and without HS+HB DMS(aq) are very low.

   *We revised the sentences (Lines 363-369) to make the point clear: However, the results from our sensitivity test do not seem to support their findings. As seen from Figure 8b, the enhancement in DMS(aq) in the Hudson Bay system (HB+HS) caused a very small change in modelled DMS(g) along the Amundsen path.*
   *The July mean difference in modelled DMS(g) mixing ratios with and without the HS+HB DMS(aq) enhancement indicates up to 500 and 250 pptv increase of DMS(g) in Hudson Strait and Hudson Bay,*

*respectively. However, the impact of the Hudson Strait and Hudson Bay system is rather locally confined during the study period (Fig. 9b), indicating either short DMS lifetime and/or inefficient transport.*

5. Lines 460-463. The size discrepancy between model and observation could be due to the effect of MSA in nucleation processes that is not considered by the model. A sentence or a very short discussion on this effect could be add here.

   *Great point. We agree that the role of MSA in formation of new particles could affect the aerosol size distribution. We added a sentence to address the concern (Lines 464-466): In addition, the size discrepancy between model and observation could be due in part to the role of MSA in the nucleation/growth of particles which is not considered in this modelling study.*

---

## Referee Comment (RC2) · Anonymous Referee #1 · 8 Jul 2019

The authors correctly answer to the questions I rise, in my opinion the paper is really excellent and it can be published on ACP.

---

## Referee Comment (RC3) · Anonymous Referee #2 · 7 Aug 2019

Ghahremaninezhad et al. present a modeling study of DMS emissions and chemistry in the Arctic using the Canadian chemical transport model GEM-MACH. The improvements in the simulations when using either satellite derived or in situ measurements of DMS(aq) will be of interest to some Atmospheric Chemistry and Physics readers. There is also an interesting and valuable discussion of the contributions of the different oxidation pathways to DMS-derived SO2 and to size-resolved sulfate aerosol concentrations. The manuscript is well written, although I have provided some minor suggestions for improving the clarity of the text below. I support publication after the following comments have been addressed.

General comments:

Defining geographic locations and regions: The manuscript is full of references to specific regions of the Canadian Arctic that would be unfamiliar to most researchers who don't work there. In addition, there are several sensitivity studies run by the authors where the emissions are modified in some of the regions but not others, and the exact boundaries of the regions where the modifications have been made are vague. Given that the actual spatial distribution of the DMS emissions is being altered, it is important that the authors provide a map indicating the major geographic regions discussed/analyzed (e.g. Hudson strait, Foxe Basin, Lancaster Sound). This addition would also make the manuscript easier to read for non-Canadian audiences.

Vertical profiles: The results shown in Figure 5 are confusing. Firstly, how is it possible to plot measurements that are below the method detection limit? By definition such data are not quantitative. Secondly, it is not clear what are the data labeled "CLIM+ave-obs", since this model run has not been defined yet in the text. (I acknowledge it is defined later.) More generally, I found this figure very hard to understand since many points are plotted at the same altitude and it is not clear which measurements and simulations correspond to the same time and location. To make the figure clearer, I would suggest binning and averaging the individual data as a function of altitude similar to what is done with the model results. Lastly, it might be helpful to plot the horizontal axis logarithmically to better show the results at lower DMS concentrations, but I will leave that to the authors' discretion.

Source Sensitivity Tests: The authors initially present a DMS dataset for the month of July and the beginning of August (Figure 7), and the DMS predictions match the measurements fairly well in August. However, only the data for July are used when the sensitivity tests are done to evaluate different emission scenarios for DMS. The authors conclude that their CLIM11+ave-Obs scenario best simulates the measurements and closes the negative bias in the base scenario (Figure 8). However, I wonder if the CLIM11+ave-Obs scenario might have a positive bias in August, since the concentrations of DMS(aq) have been increased substantially. The conclusions of the sensitivity tests would be more convincing if the entire (both July and August) measurement dataset was used, not just the part that initially showed the negative bias. I realize that extending the GEM-MACH runs might not be possible because of limitations on computational resources, but if it is possible, then I think including the August data in the comparison in Figure 8 would be very interesting and make the conclusions stronger.

Minor comments:

Abstract: The months considered in the study are a little unclear here and also seem to change throughout the text. July and August are mentioned in the first paragraph, but then in the last sentence of the second paragraph only July is named.

Line 43: Can ship emissions be a potential source of anthropogenic sulfate that is emitted in the Arctic rather than transported from southern latitudes?

Figure 1: Given that DMS(aq) doesn't exist over land surfaces, the panel showing the difference in concentrations should be white over land, similar to the panels for the SAT and CLIM11 datasets. Also, if data for August was used, then it would be potentially useful to show that data in this figure as well.

Line 200: I think there is a mistake here and it should be written than the model was run for July and August 2014, at least for some runs.

Figure 2: It would be helpful to add a lat/lon grid to the maps.

Figure 3: The height corresponding to the data that is given in the Figure caption is different from that given in the first sentence of Section 4.1. Please clarify.

Figure 6: The author's should specify the linear regression method used since that will influence the slope obtained. This comment also applies to Figure 10 as well. Also, in the figure caption, "Polar6" should have a space before the number.

Figure 7: It appears there is an error in the figure caption. It states that the CIMS

data runs to the end of July, but from the figure, there is data through the first week of August.

Figure 9: I think that the name "Lana", which I assume stands for "Lana et al.", should be replaced with the actual names of the model run as defined in Table 1 (i.e. CLIM11).

Figure 10: The legend should use run names that are consistent with Table 1.

Figure 12: What is specifically plotted in Panel B? Is the value the percentage of SO2 from DMS or the percentage increase in SO2 over the background value after DMS is implemented in the model?

Line 360: Do the authors have any have explanation why the GEOS-Chem results are different from GEM-MACH?

Line 443 – 444: I suggest using units of ng/m3 for the aerosol concentration to facilitate comparison with measurements, and also since those units are used later in the manuscript.

Line 460: It would seem pertinent to reference Croft et al. ACP 2019 here as well.

Lines 476 – 478: The wording is incorrect here. It is stated that "biogenic sulfate particles are the dominant non-sea-salt particles", however the findings of Ghahremaninezhad et al. 2016 are for non-sea-salt sulfate only and not the total particle mass. This is an important difference that should be clearly stated here as there is much new evidence from the same field study that there is an important organic contribution to arctic aerosol in the summertime (e.g. Croft et al. ACP 2019; Tremblay et al. ACP 2019; Burkart et al. GRL 2017)

Line 502: The measured concentration of 10 – 35 ng/m3 corresponds to what size range? The authors should explain why there is a range. Does it represent the measurement uncertainty or something else?

Table 1: The last sensitivity study, in which a 75% yield of SO2 from the OH-addition

pathway is considered, has been omitted from the table. I also suggest that the authors add the basic statistical metrics for each sensitivity run (e.g. mean bias, normalized mean bias, R, RMSE). Currently, these values are scattered throughout the text and not easily found.

Typographical and grammatical errors:

Line 32: Delete "the" in the first sentence.

Line 229: Atmospheric DMS(g) SAMPLES were collected. . .

Line 319: effects of the potential uncertainty IN sources. . .

Line 382: Here and elsewhere in the text, quotation marks are used occasionally with the name of the model run. I would delete them in the text, since it is not clear what purpose they serve.

Figure 11: Scientific notation should not be used in panel (d) to be consistent with the other panels.

---

## Referee Comment (RC4) · Anonymous Referee #3 · 20 Aug 2019

Review

This paper uses a NWP model with atmospheric chemistry and aerosol microphysics to compare and evaluate different DMS inventories over the Canadian Arctic. Additionally, the authors investigate the role of DMS in the simulation of Arctic aerosol. There results indicate (in general) a very poor agreement with cruise and aircraft observations of DMS. This is unsurprising given the short temporal timescale of the observations used. It was found that a satellite derived product with 8-day resolution was 'better' than a monthly climatology (Lana).

Overall the paper is very strangely structured and difficult to follow. It also contains typos and labelling mistakes which must be addressed before publication. Additionally, many of the plots need improvement. In general the paper presents some interesting ideas but requires significant editorial and scientific clarification (and potentially revision) before publication. Below are my comments to improve the paper:

Line 31: change 'The atmospheric aerosol' to 'Atmospheric aerosol' (aerosol is plural)

Line 37: Bates et al., 1987 is a extremely outdated reference, use https://journals.ametsoc.org/doi/full/10.1175/BAMS-D-15-00317.1 or https://journals.ametsoc.org/doi/pdf/10.1175/BAMS-D-14-00145.1 for most up to date state of Arctic measurement network.

Line 51: 'On the global scale, the CLAW hypothesis may be flawed,' I think you mean that the impact of the feedback is trivial.

Line 53: 'However, recent atmospheric observation and modeling studies suggest a significant role for DMS(g) in particle formation above oceans, especially in remote areas with low concentrations of pre-existing aerosol such as the Arctic Ocean in summer (Leaitch et al., 2013; Ghahremaninezhad et al., 2016; Quinn et al., 2017).'

This gives the impression that we would potentially see a CLAW feedback in the Arctic. However, greater cloud cover is more likely to warm the Arctic surface rather than cool it so the inference that 'CLAW' could occur is incorrect. Please reword.

Line 67: 'For example, the abstraction pathway (with the ratio of 75% of total OH and DMS oxidation) is the dominant reaction at 300 K (Hynes et al., 1986).'

I'm not sure why a reaction pathway that dominates at 27ïĆřC is relevant to Arctic atmospheric chemistry (even in summer).

Line 93: 'to have' please change to 'had' and 'of comparable level' to 'comparable'

Line 93: Are these seawater concentrations of DMS or atmospheric concentrations?

Line 167: I would suggest changing CLIM1 to LANA. Lana is well known climatology and immediately recognizable to modellers.

Line 189: 'coarse' should be 'coarser'

Figure 1 (and all others): Please do not use the rainbow colour scale in plots it is very difficult to interpret and distorts the results. In particular, your use of the rainbow scale for a difference plot makes interpreting the plot very hard. I suggest using a brewer colour scale or equivalent.

Line 206: 'In the case of simulation using CLIM1, constant (temporally) climatology for the month of July is used, while in the case of simulation using SAT, DMS(aq) is updated approximately every 8 days whenever the satellite-derived DMS(aq) is available'

What do mean by when they are available? Are they sometimes unavailable? Additionally, as I understand this climatology is merged with Lana at high-latitudes so regions in this model run also have static DMS concentration over the month. What percentage of this new DMS product is actually Lana?

Line 208: 'Figure S1 shows the satellite-derived DMS(aq) concentrations for the SAT time intervals, every 8 days, during July and August 2014 (July 1st to 3rd, July 4th to 11th, July 12th to 19th, July 20th to 27th, July 28th to August 4st and August 5th to 12th).'

Does this mean Figure 1 shows the average?

Figure 3: Please see my comments on Figure 1. At the moment it looks like there could be significant differences in DMS concentration over regions where the DMS climatologies are identical?

Figure 4: see comments on figures 1 and 2.

Line 260: "These flux estimates, based on measurements, are comparable with the present simulations."

With both CLIM1 and SAT? So changing the DMS inventory has had no impact on your DMS emissions?

Figure 5: This figure is difficult to understand. What do the grey dots represent? CLIM11+ave-obs is not explained in the caption or the text. Due to the linear y-axis it is very difficult to judge the fit in the model BL (which is arguably the most important region).

Line 275: 'The scatter plot in Figure 6 shows the statistical comparison of the model simulations (SAT and CLIM11) with the observation results. Overall, observation and model results are of similar magnitude, but not correlated. The simulation using SAT is in slightly better agreement with the measurement based on root mean squared error and mean bias values of 27.6 and -4.7 compared to 29.5 and -6.6 for the simulation using CLIM11, also better correlation coefficient (as shown in Fig. 6).'

I'm unsure how model and observations can be of similar magnitude but not correlated? This figure shows a terrible agreement between the model and observations-there is really no other interpretation. Additionally, you have included a regression line on what is clearly not a linear relationship and which is not statistically relevant to a model evaluation (typically you would add a one to one line to highlight agreement). Overall, it is unsurprising that the model is unable to capture aircraft point measurements even with a relatively higher (8-day) DMS resolution – which is likely the reason SAT is a slightly better fit. This looks like a clear example of sample bias and I question the usefulness of this comparison.

Line 302: 'These are the physical parameters affecting the sea-air flux. Overall the model is in good agreement with observations, given the model resolution,'

Really? That is not how I would interpret this plot. The most significant differences, particularly towards the end of the month, seem to coincide with the models failure to simulate sea surface temperatures. I'm also unsure what resolution you are referring to here, spatial or temporal?

Line 329: You refer to a figure 8c but figure 8 is not labelled as such.

Figure 9: Again a rainbow colour bar is a bad choice in general but for a difference plot doubly so. Additionally, the plots are also labelled in the caption (a,b,c) but not in the plot.

Line 344 (section 4.2.1):" indicating that the melt pond sources did not contribute to the two high DMS(g) events observed onboard the Amundsen."

This simulation is interesting however it is unclear which DMS climatology that you have used in this simulation, as static monthly climatology (which is very unlikely to capture specific plume events) or the 8-day resolution SAT inventory. The use of either is unlikely to reproduce specific DMS events observed on 2-3 day timescale (particularly Lana) – which I would argue is the reason for your poor model evaluations. Therefore, it seems a stretch to rule out melt ponds as an important DMS source. Overall, this section seems entirely divorced from any other part of the paper.

Section 4.2.3: In the simulation CLIM1-ave+Obs you have (as I understand it) used observations from the NETCARE campaign to update the Lana climatologies. Why update Lana, when SAT has a higher temporal resolution and you have previously shown that SAT is better (i.e. Fig 10)?

Line 380 (Fig 10): 'The statistical evaluations in this figure indicates a significant improvement in CLIM 11 model-observation comparison with this update (Fig. 10).'

Really? I don't see a significant improvement. Correlation between observations and the model is low for all three climatologies, although SAT is the best (again I would argue because it has a higher temporal resolution), which begs the question why it wasn't used for the update. Additionally, as in the previous figure, you have included a regression line rather than a one-to-one line which would make it easier to judge the comparison.

Section 4.3: Throughout this section is unclear exactly what simulations you are comparing. It seems you have switched from comparing DMS climatologies to comparing models with and without DMS. To make the paper flow better I would suggest beginning the results section with the impact of adding DMS to the model and then discussing the impact of different DMS inventories.

Line 458: 'In general, GEM-MACH suggests the enhancement of particles between 50 to 100 nm to be higher than particles between 10 to 50 nm for the high Arctic. This difference between Abbatt et al., (2019) and GEM-MACH results could be partly due to missing other natural sources (e.g., organics, see Burkart et al., 2017; Willis et al, 2016) in the model. Possible inadequacy in model representation of particle nucleation process may also contribute to the size discrepancy between model and observation.'

The enhancement of larger particles is more likely the result of too larger a condensation sink leading to condensation of SO2 rather than new particle formation. This could result from an underestimation of sink processes or an overestimation of other aerosol sources.

Line 471: 'The model simulation in this study compares well with the observations'

I'd say reasonable well, given the number of observations. However, at least 20% of the time the model is almost a factor of 10 lower than the observations. Why is the model so wrong on July 14th?

Line 529: 'By adding DMS(g) in the GEM-MACH model, the atmospheric SO2 concentration increased (up to ~100% for some regions). This increase in may play a significant role in the growth and nucleation of aerosols.'

Doe this improve the models representation of SO2?

---

## Author Response (AR1)

We are very thankful for the reviewers' comments. We believe that we have answered all the comments/concerns. The revised version is attached and the modified texts in the manuscript are highlighted. Please see below; a point-to-point response to the comments (in *italic font*) from each of the three reviewers.

**Reviewer # 1**

1. Lines 40-42. Among the sources of sulfate, the volcanic is not reported here. In north hemisphere, and especially in Iceland, the presence of volcanoes and fumaroles can furnish a contribution to SO2 budget.

   *Yes, volcanoes and fumaroles do contribute to global atmospheric sulfur budget. It is believed that they contribute to 5 – 7 % of global total sulfur emissions. In the manuscript, we had mentioned the main sources during the Arctic summer. However, we have now added the geological sources as well, to make a general statement. Here are the modifications to address this comment:* (Line 41-42): Sulfate aerosols in the Arctic atmosphere originate from the anthropogenic, sea salt, geological and biogenic sources. (Lines 47-48): The geological sources include $SO_2$ emission from volcanoes and Smoking Hills (Yang et al., 2018; Rempillo et al., 2011).

2. *Line 204. CLIM11 instead of CLIM1*

   *Thanks, we fixed that.*

3. *Lines 229-232. Here the sampling height (of mean height) of sampling has to be reported. As reported below in the paper, this information is useful as the DMS concentration depend to the height of sampling.*

   *Great point. We agree that the sampling heights need to be mentioned, since DMS concentrations change with the altitude. We added the sampling heights: (Line 226):* 50 to 5000 m above the mean sea level. *Please note, as mentioned in the manuscript, more details about the observation methods/sampling heights are reported in Ghahremaninezhad et al., 2017.*

4. *Lines 364-365. This sentence is not clear, what is clear from figure 8b is that the difference between DMS results from CLIM11 with and without HS+HB DMS(aq) are very low.*

   *We revised the sentences (Lines 359-368) to make the point clear:* However, the results from our sensitivity test do not seem to support their findings. As seen from Figure 8b, the enhancement in DMS(aq) in the Hudson Bay system (HB+HS) caused a very small change in modelled DMS(g) along the Amundsen path. One possible reason for the difference in model results between GEM-MACH and GEOS-Chem could be in model resolutions (e.g. GEOS-Chem model, 2 × 2.5° and GEM-MACH, 15 km).
   The difference in modelled July-averaged DMS(g) mixing ratios with and without the HS+HB DMS(aq) enhancement, shown in Figure 9b, indicates that the impact of the Hudson Strait and Hudson Bay system is rather locally confined during the study period. The HS+HB DMS(aq) enhancement led to an increase in modelled DMS(g) of up to 300 pptv in the Hudson Strait area but the area of increase does not extend much beyond the area of DMS(aq) enhancement, indicating either short DMS(g) lifetime and/or inefficient transport in this case.

5. Lines 460-463. The size discrepancy between model and observation could be due to the effect of MSA in nucleation processes that is not considered by the model. A sentence or a very short discussion on this effect could be add here.

*Great point. We agree that the role of MSA in formation of new particles could affect the aerosol size distribution. We added a sentence to address the concern (Lines 464-465): In addition, the size discrepancy between model and observation could be due in part to the role of MSA in the nucleation/growth of particles*

**Reviewer # 2**

General Comments:

1. Defining geographic locations and regions: The manuscript is full of references to specific regions of the Canadian Arctic that would be unfamiliar to most researchers who don't work there. In addition, there are several sensitivity studies run by the authors where the emissions are modified in some of the regions but not others, and the exact boundaries of the regions where the modifications have been made are vague. Given that the actual spatial distribution of the DMS emissions is being altered, it is important that the authors provide a map indicating the major geographic regions discussed/analyzed (e.g. Hudson strait, Foxe Basin, Lancaster Sound). This addition would also make the manuscript easier to read for non-Canadian audiences.

   *In Fig. 2, we added the latitude/longitude information and specified locations (such as Hudson strait, Foxe Basin, Lancaster Sound). Also, we added Fig. S3 to show the locations with the updated CLIM11 DMS(aq) values (Line 320).*

2. Vertical profiles: The results shown in Figure 5 are confusing. Firstly, how is it possible to plot measurements that are below the method detection limit? By definition such data are not quantitative. Secondly, it is not clear what are the data labeled "CLIM+aveobs", since this model run has not been defined yet in the text. (I acknowledge it is defined later.) More generally, I found this figure very hard to understand since many points are plotted at the same altitude and it is not clear which measurements and simulations correspond to the same time and location. To make the figure clearer, I would suggest binning and averaging the individual data as a function of altitude similar to what is done with the model results. Lastly, it might be helpful to plot the horizontal axis logarithmically to better show the results at lower DMS concentrations, but I will leave that to the authors' discretion.

   *We have made Fig. 5 clear now. We have removed the BDL data (Thanks for the comment. It was a mistake: "below the precision of analysis" instead of "below detection limit"). Also, we added a sentence in the text (Line 261) and in caption of this figure to clarify that CLIM11+ave-Obs simulation is defined in section 4.2.3. In Fig.5, the lines are the vertical profiles of DMS(g) from the model simulations, and the dots are atmospheric DMS (pptv) from the model simulations extracted along the Polar 6 flight path coincide with observation data (GC-SCD). There are just few measurement data available for each flight and it is not possible to bin and plot the vertical profile for the observation results. We have also changed the horizontal axis to logarithmic scale as per reviewer's suggestion.*

3. Source Sensitivity Tests: The authors initially present a DMS dataset for the month of July and the beginning of August (Figure 7), and the DMS predictions match the measurements fairly well in August. However, only the data for July are used when the sensitivity tests are done to evaluate different emission scenarios for DMS. The authors conclude that their CLIM11+ave-Obs scenario best simulates the measurements and closes the negative bias in the base scenario (Figure 8). However, I wonder if the CLIM11+ave-Obs scenario might have a positive bias in August, since

the concentrations of DMS(aq)have been increased substantially. The conclusions of the sensitivity tests would be more convincing if the entire (both July and August) measurement dataset was used, not just the part that initially showed the negative bias. I realize that extending the GEM-MACH runs might not be possible because of limitations on computational resources, but if it is possible, then I think including the August data in the comparison in Figure 8 would be very interesting and make the conclusions stronger.

*The initial sensitivity tests were carried out for July only as the majority of the observations were in July. We have now extended the Clim11+ave-Obs simulation to cover the entire study period and included the results in the revised manuscript (see revised Figure 8).*

Minor comments:

1. Abstract: The months considered in the study are a little unclear here and also seem to change throughout the text. July and August are mentioned in the first paragraph, but then in the last sentence of the second paragraph only July is named.

   *We changed the last sentence of the second paragraph in the Abstract: Summer time instead of only July.*

2. Line 43: Can ship emissions be a potential source of anthropogenic sulfate that is emitted in the Arctic rather than transported from southern latitudes?

   *We added a sentence in lines 44-45 to address this comment: Ship emissions, particularly in summer time, contribute to the anthropogenic sulfate in the Arctic as well (e.g., Gong et al., 2018a).*

3. Figure 1: Given that DMS(aq) doesn't exist over land surfaces, the panel showing the difference in concentrations should be white over land, similar to the panels for the SAT and CLIM11 datasets. Also, if data for August was used, then it would be potentially useful to show that data in this figure as well.

   *We changed the colour scale to address this comment (see revised Fig. 1, lower panel). Also, we added CLIM11 August averaged DMS(aq) in Figure S2 (Line 181).*

4. Line 200: I think there is a mistake here and it should be written than the model was run for July and August 2014, at least for some runs.

   *Thanks, we have clarified this in the revised version (Line 197: The simulation was carried out for July 1st to August 8th of 2014 in this study).*

5. Figure 2: It would be helpful to add a lat/lon grid to the maps.

   *Yes, we added lat/long grid to the map (Fig. 2).*

6. Figure 3: The height corresponding to the data that is given in the Figure caption is different from that given in the first sentence of Section 4.1. Please clarify.

   *Thanks, the height in the caption of Fig. 3 should be 20 m, the lowest model level, (as mentioned in the section 4.1).*

7.  Figure 6: The author's should specify the linear regression method used since that will influence the slope obtained. This comment also applies to Figure 10 as well. Also, in the figure caption, "Polar6" should have a space before the number.

    *We have now specified linear regression in the caption of the figures 6 and 10. Also, we fixed the typo (Polar 6) in the Fig. 6 caption.*

8.  Figure 7: It appears there is an error in the figure caption. It states that the CIMS data runs to the end of July, but from the figure, there is data through the first week of August.

    *We have fixed the error in the figure caption (see revised Figure 7 caption).*

9.  Figure 9: I think that the name "Lana", which I assume stands for "Lana et al.", should be replaced with the actual names of the model run as defined in Table 1 (i.e. CLIM11).

    *We have fixed that in the figure cation (see revised Figure 9 caption).*

10. Figure 10: The legend should use run names that are consistent with Table 1.

    *Done. See revised Figure 10.*

11. Figure 12: What is specifically plotted in Panel B? Is the value the percentage of SO2 from DMS or the percentage increase in SO2 over the background value after DMS is implemented in the model?

    *It is the relative percentage increment to the modelled $SO_2$ (lower panel) during July 2014 due to adding DMS(g) in GEM-MACH. We changed the caption of the fig.12: Figure 12: The absolute increment (upper panel) and relative (percentage) increment (lower panel) to the modelled July-averaged $SO_2$ concentration due to adding DMS(g) in GEM-MACH.*

12. Line 360: Do the authors have any have explanation why the GEOS-Chem results are different from GEM-MACH?

    *We added the following on one possible reason: Lines 362-363: One possible reason for the difference in model results between GEM-MACH and GEOS-CHEM could be in model resolutions (e.g. GEOS-Chem model, $2 \times 2.5°$ and GEM-MACH, 15 km).*

13. Line 443 – 444: I suggest using units of ng/m3 for the aerosol concentration to facilitate comparison with measurements, and also since those units are used later in the manuscript.

    *We have added the conversion to ng/m$^3$(see line 443-445 in the revised manuscript).*

14. Line 460: It would seem pertinent to reference Croft et al. ACP 2019 here as well.

    *Thanks for the suggestion, we have added the reference.*

15. Lines 476 – 478: The wording is incorrect here. It is stated that "biogenic sulfate particles are the dominant non-sea-salt particles", however the findings of Ghahremaninezhad et al. 2016 are for non-sea-salt sulfate only and not the total particle mass. This is an important difference that should be clearly stated here as there is much new evidence from the same field study that there is an important organic contribution to arctic aerosol in the summertime (e.g. Croft et al. ACP 2019; Tremblay et al. ACP 2019; Burkart et al. GRL 2017).

*Yes, we have fixed this: Lines 478-480: Also, Ghahremaninezhad et al., (2016) found that biogenic sulfate particles are the dominant non-sea-salt sulfate particles during July in the Arctic atmosphere, and > 63 % of non-sea-salt sulfate fine aerosol (<0.49 μm) were from biogenic source (DMS).*

16. Line 502: The measured concentration of $10 - 35$ ng/m3 corresponds to what size range? The authors should explain why there is a range. Does it represent the measurement uncertainty or something else?

    *We have specified the size ranges: Lines 502-505: However, assuming that sulfate comprises about 40% of the submicron particles in the clean air at Alert (Leaitch et al., 2018), the sulfate mass concentrations are estimated in the range of, at least, 10 ng/m$^3$ for particle size range of 20-100 nm, and 35 ng/m$^3$ for particle size range of 20-200 nm at Alert from natural sources.*

17. Table 1: The last sensitivity study, in which a 75% yield of SO2 from the OH-addition pathway is considered, has been omitted from the table. I. also suggest that the authors add the basic statistical metrics for each sensitivity run (e.g. mean bias, normalized mean bias, R, RMSE). Currently, these values are scattered throughout the text and not easily found.

    *We have added the run considering 75% yield of SO$_2$ from OH-addition in the table 1. We have summarized the basic statistical metrics in Table 2.*

    Typographical and grammatical errors:

18. Line 32: Delete "the" in the first sentence.

19. Line 229: Atmospheric DMS(g) SAMPLES were collected. . .

20. Line 319: effects of the potential uncertainty IN sources. . .

21. Line 382: Here and elsewhere in the text, quotation marks are used occasionally with the name of the model run. I would delete them in the text, since it is not clear what purpose they serve.

22. Figure 11: Scientific notation should not be used in panel (d) to be consistent with the other panels.

    *Thanks, we fixed the typographical and grammatical errors.*

**Reviewer # 3**

1. Line 31: change 'The atmospheric aerosol' to 'Atmospheric aerosol' (aerosol is plural).

   *We fixed that.*

2. Line 37: Bates et al., 1987 is a extremely outdated reference, use https://journals.ametsoc.org/doi/full/10.1175/BAMS-D-15-00317.1 or https://journals.ametsoc.org/doi/pdf/10.1175/BAMS-D-14-00145.1 for most up to date state of Arctic measurement network.

*We added the references.*

3. Line 51: 'On the global scale, the CLAW hypothesis may be flawed,' I think you mean that the impact of the feedback is trivial.

*The statement was referring to some of the recent discussions on CLAW hypothesis as detailed in Quinn and Bates (2011). Though the validity and realization of the CLAW hypothesis are still debated, a general consensus may be that the CLAW hypothesis is an oversimplification of a highly complicated ocean-atmosphere system. Since this study focuses on the impact of DMS on atmospheric aerosol formation and growth in the summer Arctic and not on its potential role in the climate system, we have removed the brief discussion on the CLAW hypothesis from the manuscript.*

4. Line 53: 'However, recent atmospheric observation and modeling studies suggest a significant role for DMS(g) in particle formation above oceans, especially in remote areas with low concentrations of pre-existing aerosol such as the Arctic Ocean in summer (Leaitch et al., 2013; Ghahremaninezhad et al., 2016; Quinn et al., 2017).' This gives the impression that we would potentially see a CLAW feedback in the Arctic. However, greater cloud cover is more likely to warm the Arctic surface rather than cool it so the inference that 'CLAW' could occur is incorrect. Please reword.

*This statement is unrelated to the CLAW hypothesis; rather it refers to the role of DMS in particle formation above ocean in remote areas suggested by recent studies. We have combined this with the discussion on DMS role in particle formation and growth and CCN later in the introduction (see lines 77-80).*

5. Line 67: 'For example, the abstraction pathway (with the ratio of 75% of total OH and DMS oxidation) is the dominant reaction at 300 K (Hynes et al., 1986).' I'm not sure why a reaction pathway that dominates at 27ï´CˇrC is relevant to Arctic atmospheric chemistry (even in summer).

*As we mentioned in the manuscript, it is just an example to say that abstraction pathway is temperature dependent. In lower temperatures the ratio is smaller but not zero, and this pathway may still be important (as indicated from our results in Figure 11b).*

6. Line 93: 'to have' please change to 'had' and 'of comparable level' to 'comparable'.

*We fixed them.*

7. Line 93: Are these seawater concentrations of DMS or atmospheric concentrations?

*DMS(aq); we clarified this in the revised manuscript.*

8. Line 167: I would suggest changing CLIM1 to LANA. Lana is well known climatology and immediately recognizable to modellers.

*In the method section, in line 165, we mentioned that we refer to Lana et al. 2011's climatology as CLIM11 (Lana et al., 2011; hereafter referred to as CLIM11). We believe it is very clear.*

9. Line 189: 'coarse' should be 'coarser'.

*We changed 'coarse' to 'coarser'.*

10. Figure 1 (and all others): Please do not use the rainbow colour scale in plots it is very difficult to interpret and distorts the results. In particular, your use of the rainbow scale for a difference plot makes interpreting the plot very hard. I suggest using a brewer colour scale or equivalent.

*We have changed the colour palate for the difference plot (from rainbow to blue-white-red) to distinguish it from the concentration plots, as suggested by this reviewer and reviewer #2. However, we have retained the rainbow colour palate for the concentration plots as we think that the different colour tones in the rainbow colour scale allow us to show the details in the concentration field better.*

11. Line 206: 'In the case of simulation using CLIM1, constant (temporally) climatology for the month of July is used, while in the case of simulation using SAT, DMS(aq) is updated approximately every 8 days whenever the satellite-derived DMS(aq) is available' What do mean by when they are available? Are they sometimes unavailable? Additionally, as I understand this climatology is merged with Lana at high-latitudes so regions in this model run also have static DMS concentration over the month. What percentage of this new DMS product is actually Lana?

*Here, we did not refer to the replacing of SAT data-set with CLIM11, and "whenever the satellite-derived DMS(aq) is available" referred to the 8-day resolution of SAT. We removed this sentence. The specific time intervals for the satellite-derived DMS(aq) are given in the text (lines 206-207 of the revised manuscript).*
*Just to clarify, the satellite-derived DMS(aq) data used in SAT run are not climatology; they are based on "real-time" satellite observation and available at a 8-day interval. As mentioned in section 2.2 (line 190-191, over the central Arctic Ocean DMS(aq) concentrations are not available from the SAT dataset due to the limitation of satellite detection in presence of sea ice. As a result, for the SAT run DMS(aq) concentrations over this region were filled in with DMS(aq) values from CLIM11.*

12. Line 208: 'Figure S1 shows the satellite-derived DMS(aq) concentrations for the SAT time intervals, every 8 days, during July and August 2014 (July 1st to 3rd, July 4th to 11th, July 12th to 19th, July 20th to 27th, July 28th to August 4st and August 5th to 12th).' Does this mean Figure 1 shows the average?

*Yes*

13. Figure 3: Please see my comments on Figure 1. At the moment it looks like there could be significant differences in DMS concentration over regions where the DMS climatologies are identical?

*In general, DMS(g) concentration values reflect the DMS(aq) concentration patterns. However, there are other factors (e.g. advection) affecting the DMS(g) concentration values.*

14. Figure 4: see comments on figures 1 and 2.

*See response on comments on figures 1.*

15. Line 260: "These flux estimates, based on measurements, are comparable with the present simulations." With both CLIM1 and SAT? So changing the DMS inventory has had no impact on your DMS emissions.

*Here we are comparing the modelled DMS fluxes with existing estimates from available observations to see if they fall within the same range of magnitudes. As shown in Figure 4 (and the discussion in 4.1, line 252-255), the differences in modelled DMS fluxes between CLIM11 and SAT are mostly in spatial distribution reflecting the differences in DMS(aq) between the two datasets.*

16. Figure 5: This figure is difficult to understand. What do the grey dots represent? CLIM11+ave-obs is not explained in the caption or the text. Due to the linear y-axis it is very difficult to judge the fit in the model BL (which is arguably the most important region).

*We have made Fig. 5 clearer now. We added a sentence in the text (Line 261) and caption of this figure to clarify that CLIM11+ave-Obs simulation is defined in section 4.2.3. In Fig.5, the lines are the vertical profiles of DMS(g) from the model simulations, and the dots are Atmospheric DMS (pptv) from the model simulations extracted along the Polar 6 flight path coincide with observation data (GC-SCD). We have also changed the horizontal axis to logarithmic scale as per reviewer # 2's suggestion.*

17. Line 275: 'The scatter plot in Figure 6 shows the statistical comparison of the model simulations (SAT and CLIM11) with the observation results. Overall, observation and model results are of similar magnitude, but not correlated. The simulation using SAT is in slightly better agreement with the measurement based on root mean squared error and mean bias values of 27.6 and -4.7 compared to 29.5 and -6.6 for the simulation using CLIM11, also better correlation coefficient (as shown in Fig. 6). I'm unsure how model and observations can be of similar magnitude but not correlated? This figure shows a terrible agreement between the model and observations-there is really no other interpretation. Additionally, you have included a regression line on what is clearly not a linear relationship and which is not statistically relevant to a model evaluation (typically you would add a one to one line to highlight agreement). Overall, it is unsurprising that the model is unable to capture aircraft point measurements even with a relatively higher (8-day) DMS resolution – which is likely the reason SAT is a slightly better fit. This looks like a clear example of sample bias and I question the usefulness of this comparison.

*We have removed the statement "but uncorrelated". The correlation is low but not unexpected as pointed out by the reviewer, i.e., aircraft point measurements vs. model simulation (given the temporal and spatial*

*resolution and the DMS(aq) available). The reviewer expects to see better results from SAT than CLIM11, because of higher SAT DMS(aq) resolution, and the statistical evaluation results shown here does support this.*

18. Line 302: 'These are the physical parameters affecting the sea-air flux. Overall the model is in good agreement with observations, given the model resolution,' Really? That is not how I would interpret this plot. The most significant differences, particularly towards the end of the month, seem to coincide with the models failure to simulate sea surface temperatures. I'm also unsure what resolution you are referring to here, spatial or temporal?

*We are referring to both temporal and spatial resolutions. We see a phase shift in the sea-surface temperature in the end of the month (between model and observation). But the model and observation are still comparable in magnitudes (the difference is likely driven by the discrepancy in sea ice cover between the analysis used in the model and the real world). By considering the flux equation, the DMS source in water has more effect on the DMS emission and is still the main source of uncertainty.*

19. Line 329: You refer to a figure 8c but figure 8 is not labelled as such. Figure 9: Again a rainbow colour bar is a bad choice in general but for a difference plot doubly so. Additionally, the plots are also labelled in the caption (a,b,c) but not in the plot.

*We added labels a, b. c to figures 8 and 9.*

20. Line 344 (section 4.2.1):" indicating that the melt pond sources did not contribute to the two high DMS(g) events observed onboard the Amundsen." This simulation is interesting however it is unclear which DMS climatology that you have used in this simulation, as static monthly climatology (which is very unlikely to capture specific plume events) or the 8-day resolution SAT inventory. The use of either is unlikely to reproduce specific DMS events observed on 2-3 day timescale (particularly Lana) – which I would argue is the reason for your poor model evaluations. Therefore, it seems a stretch to rule out melt ponds as an important DMS source. Overall, this section seems entirely divorced from any other part of the paper.

*The sensitivity testes discussed in section 4.2 are focused on possible sources contributing to the model under-prediction of atmospheric DMS in comparison to the observations aboard Amundsen cruise through the Canadian archipelago, particularly the high DMS event observed during July 18 – 21, 2014. As we mentioned in the manuscript, the no-Ice test is an extreme case for considering the potential contribution from melt ponds, by assuming that the entire ice-covered portion of the Arctic ocean is covered by melt ponds, and the DMS(aq) concentrations in these melt ponds are the same as open water sea surface water DMS(aq) concentration, and the flux exchange is the same as over open water. Amundsen covered a relatively large area in the 2-3 day interval with high DMS(g) concentrations.*

21. Section 4.2.3: In the simulation CLIM1-ave+Obs you have (as I understand it) used observations from the NETCARE campaign to update the Lana climatologies. Why update Lana, when SAT has a higher temporal resolution and you have previously shown that SAT is better (i.e. Fig 10)?

*Because CLIM11 is based on few measurements in the Arctic, and it can be a significant source of uncertainty. To address this uncertainty, we replaced CLIM11 with the observed DMS(aq) data. Also, since CLIM11 is broadly used by other modelling groups in existing publications, we decided to use CLIM11 as the base case for the sensitivity tests.*

22. Line 380 (Fig 10): 'The statistical evaluations in this figure indicates a significant improvement in CLIM 11 model-observation comparison with this update (Fig. 10).' Really? I don't see a significant improvement. Correlation between observations and the model is low for all three climatologies, although SAT is the best (again I would argue because it has a higher temporal resolution), which begs the question why it wasn't used for the update. Additionally, as in the previous figure, you have included a regression line rather than a one-to-one line which would make it easier to judge the comparison.

*The improvement in model-observation comparison from the CLIM11+ave-Obs run is clearly shown in Figure 10. We have added the one-to-one line in both Figure 10 and Figure 6 as suggested by the reviewer (see the answer for the previous question too).*

23. Section 4.3: Throughout this section is unclear exactly what simulations you are comparing. It seems you have switched from comparing DMS climatologies to comparing models with and without DMS. To make the paper flow better I would suggest beginning the results section with the impact of adding DMS to the model and then discussing the impact of different DMS inventories.

*We have stated in the very beginning of   section 4.3 that the discussions in this section are based on the simulation results from the CLIM11+ave-Obs run (see Lin 387). The main goal of the research was the implementation of DMS in GEM-MACH in order to investigate the impact of DMS on aerosols in the Arctic summer. We need to first evaluate modelled DMS before we can discuss the impact of DMS based the model results. The rearrangement suggested by this reviewer, i.e., discussing the model results on DMS impact before evaluating modelled DMS does not seem logical in our opinion.*

24. Line 458: 'In general, GEM-MACH suggests the enhancement of particles between 50 to 100 nm to be higher than particles between 10 to 50 nm for the high Arctic. This difference between Abbatt et al., (2019) and GEM-MACH results could be partly due to missing other natural sources (e.g., organics, see Burkart et al., 2017; Willis et al, 2016) in the model. Possible inadequacy in model representation of particle nucleation process may also contribute to the size discrepancy between model and observation.' The enhancement of larger particles is more likely the result of too larger a condensation sink leading to condensation of SO2 rather than new particle formation. This could result from an underestimation of sink processes or an overestimation of other aerosol sources.

*It is true that too large a condensation sink in the model would lead to more sulfate condensing on existing aerosols than nucleating and hence more enhancement in larger sizes. At the same time, too large a condensation sink in the model would imply a model over-prediction of existing aerosols. However, in a separate study (manuscript in preparation currently) we show that the model (GEM-MACH without DMS) generally under-predicted aerosols over the study area (based on comparison with in-situ measurement), which led us to believe that the inadequacy in representing particle nucleation in the model (see discussion towards the end of this paragraph) is more likely one of the causes. Nonetheless we can not rule out the possible over prediction of condensation sink definitively and we have added this being a possibility in the revised version.*

*Line 458-465: In general, GEM-MACH suggests the enhancement of particles between 50 to 100 nm to be higher than particles between 10 to 50 nm for the high Arctic. This difference between Abbatt et al., (2019) and GEM-MACH results could be partly due to missing other natural sources (e.g., organics, see Croft et al., 2019, Burkart et al., 2017; Willis et al, 2016) in the model. Possible inadequacy in model representation of particle nucleation process and over-prediction of condensation sink may also contribute to the size discrepancy between model and observation. For example, in the model new particles formed through nucleation are added to the first model size bin (10 – 20 nm), at sizes considerably bigger than nucleating particles in the real world (e.g., Kulmala et al., 2006). In addition, the size discrepancy between model and observation could be due in part to the role of MSA in the nucleation/growth of particles which is not considered for this modelling study.*

25. Line 471: 'The model simulation in this study compares well with the observations'**.** I'd say reasonable well, given the number of observations. However, at least 20% of the time the model is almost a factor of 10 lower than the observations. Why is the model so wrong on July 14th?

*The discussion is referring to a comparison of modelled and observed biogenic sulfate in three size ranges on the Amundsen cruise. As shown in Figure 15, the agreement is good particularly at higher latitudes (July 16 and later). We have reworded the sentence accordingly (Lines 473-475: At higher latitudes (July 16th and later), the model simulation in this study compares well with the observations and also demonstrates that a larger fraction of DMS derived sulfate (or biogenic sulfate) is found in aerosols with sizes < 0.49 μm).*

26. Line 529: 'By adding DMS(g) in the GEM-MACH model, the atmospheric SO2 concentration increased (up to _100% for some regions). This increase in may play a significant role in the growth and nucleation of aerosols.' Does this improve the models representation of SO2?

*We agree that it would be good to look at if the implementation of DMS leads to an improved model prediction of $SO_2$ in the Arctic. However, there is a lack of reliable observational data on $SO_2$ concentrations in the summer Arctic (most of the conventional $SO_2$ instruments do not have the precision required to make reliable measurement at low concentration levels found in the summer Arctic). We do not have $SO_2$ measurements for such evaluation.*

---

## Author Response (AR2)

We would like to thank Referee#2 for the suggestion. Please see below; a point-by-point response to the comment (in *italic font*) and the revised version. The modified texts in the manuscript are highlighted.

**Report #1, Referee #2:**

I thank the authors for having incorporated my comments and I think that the manuscript is now suitable for publication in ACP. However, I have one final suggestion for revision before publication.

On lines 464-465, the authors suggest that the size discrepancy between the model and the observations could be due to "the role of MSA in the nucleation/growth of particles". However, recent field measurement studies have found that while MSA correlates with particle growth in the Arctic, it is not a major contributor to the mass of Aitken mode particles or their growth (Willis et al. Atmos. Chem. Phys. 2016, doi.org/10.5194/acp-16-7663-2016 and Tremblay et al. Atmos. Chem. Phys. 2019, doi.org/10.5194/acp-19-5589-2019). Furthermore, using the GEOS-Chem model, Croft et al. were not able to reproduce observed size distributions in the Arctic even when particle growth by MSA condensation was included in the model (Croft et al. Atmos. Chem. Phys. 2019, doi.org/10.5194/acp-19-2787-2019). The manuscript could include some speculation that MSA (or lack thereof) might explain the size discrepancy, since there still needs to be additional studies on this topic, but the previous work that suggests otherwise should be referenced and discussed.

Great point. We included the previous observation and modeling studies to address the concern. (Lines 463-468): However, recent field measurement studies have found that while MSA correlates with particle growth in the Arctic, it is not a major contributor to the mass of Aitken mode particles or their growth (Willis et al., 2016; Tremblay et al., 2019). Furthermore, using the GEOS-Chem model, Croft et al. (2019) found that the condensation of MSA dose not lead to sufficient particle growth and they were not able to reproduce observed size distributions in the Arctic even when particle growth by MSA condensation was included in the model. Further studies are needed to quantify the contribution of MSA in nucleation/growth of aerosol in the Arctic atmosphere.

**Dimethyl sulfide and its role in aerosol formation and growth in the Arctic summer – a modelling study**

Roghayeh Ghahremaninezhad1, Wanmin Gong1, Martí Galí2, Ann-Lise Norman3, Stephen R. Beagley1, Ayodeji Akingunola1, Qiong Zheng1, Alexandru Lupu1, Martine Lizotte2, Maurice Levasseur2, W. Richard Leaitch1.

[revised manuscript text omitted]